# Beyond Last-Click: An Optimal Mechanism for Ad Attribution

**Nan An**
Gaoling School of Artificial Intelligence
Renmin University of China
Beijing, China
annan0425@ruc.edu.cn

**Weian Li**
School of Software
Shandong University
Jinan, China
weian.li@sdu.edu.cn

**Qi Qi**[*]
Gaoling School of Artificial Intelligence
Renmin University of China
Beijing, China
qi.qi@ruc.edu.cn

**Changyuan Yu**
Baidu Inc.
Beijing, China
yuchangyuan@baidu.com

**Liang Zhang**
Gaoling School of Artificial Intelligence
Renmin University of China
Beijing, China
zhang.liang@ruc.edu.cn

## Abstract

Accurate attribution for multiple platforms is critical for evaluating performance-based advertising. However, existing attribution methods rely heavily on the heuristic methods, e.g., Last-Click Mechanism (LCM) which always allocates the attribution to the platform with the latest report, lacking theoretical guarantees for attribution accuracy. In this work, we propose a novel theoretical model for the advertising attribution problem, in which we aim to design the optimal dominant strategy incentive compatible (DSIC) mechanisms and evaluate their performance. We first show that LCM is not DSIC and performs poorly in terms of accuracy and fairness. To address this limitation, we introduce the Peer-Validated Mechanism (PVM), a DSIC mechanism in which a platform's attribution depends solely on the reports of other platforms. We then examine the accuracy of PVM across both homogeneous and heterogeneous settings, and provide provable accuracy bounds for each case. Notably, we show that PVM is the optimal DSIC mechanism in the homogeneous setting. Finally, numerical experiments are conducted to show that PVM consistently outperforms LCM in terms of attribution accuracy and fairness.

## 1 Introduction

Online advertising has become the dominant force in the global advertising landscape, with expenditures projected to exceed $790 billion in 2024—accounting for over 72% of total ad spend—and continuing to grow at a consistent rate of more than 10% annually. This substantial and growing capital investment calls for the development and application of robust methodologies to optimize budget allocation across diverse digital platforms.

---

[*]Corresponding author.

39th Conference on Neural Information Processing Systems (NeurIPS 2025).

Advertising attribution, the process of assigning credit for user conversions (such as app downloads or product purchases) to the platforms that contributed to them, plays a central role in guiding these allocation decisions and has consequently garnered significant attention. In practice, attribution reflects a variety of design principles and business objectives. Methods range from simple heuristics such as first-click and time-decay attribution to data-driven approaches based on machine learning and causal inference. Among these, last-click attribution has become the industry default due to its simplicity and its practical relevance for measuring revenue-driven, bottom-of-funnel conversions.

Under last-click attribution, the platform that most recently interacted with the user receives full conversion credit. Because these credits directly determine performance metrics and future budget allocation, platforms have a strong incentive to manipulate the timing of their reports to appear last in the user's interaction sequence. Such strategic behavior can distort attribution outcomes, overstating the influence of certain platforms even on its own terms.

This manipulation has become increasingly feasible in modern advertising ecosystems, especially when the advertiser does not control the landing page—such as app installations through app stores or purchases on major e-commerce platforms—where click events cannot be directly measured. In the past, advertisers relied on redirect-based tracking flows, where an intermediary measurement partner (MMP) logged the click before redirecting the user to the final landing page, thus providing an independent, verifiable timestamp. However, the industry has since shifted toward *redirect-less* tracking paradigm, in which user navigation and click reporting are decoupled to improve latency and privacy. Without an intermediary verifier, advertisers now depend entirely on platform's self-reported timestamps, making strategically timed reports both feasible and practically undetectable.[2]

Nevertheless, such strategic behavior has received limited attention in the academic literature. Most prior work on advertising attribution instead focuses on modeling platform contributions to conversions using increasingly sophisticated statistical or machine learning methods, under the assumption that platforms passively and truthfully report user interaction data. In this paper, we initiate the study of advertising attribution from a mechanism design perspective, treating platforms as strategic agents that may misreport in order to maximize their assigned credit. Rather than proposing a new attribution philosophy, we work within the prevailing logic of last-click attribution and ask: *How can we design an attribution mechanism such that platforms have no incentive to misreport, while still assigning credit to the platform with the true last click?*

**Main Contribution**   To address the above question, we first model the advertising attribution scenario as a game-theoretic model in which multiple platforms strategically submit user interaction logs to compete for conversion credit. The advertiser then allocates credit according to a predefined attribution rule. In this model, our analysis focuses on characterizing dominant strategy incentive compatible (DSIC) mechanisms, and evaluating the performance of different attribution mechanisms, using two key metrics: accuracy and fairness. Accuracy measures the alignment between the assigned and true contributors, while fairness assesses whether each platform receives its deserved share of credit in expectation. Detailed results are presented in Table 1, with proofs in the full version.

Table 1: Mechanism performance under different settings

| | DSIC | Fair | Accuracy (Homogeneous) | | Accuracy (Heterogeneous) | |
|---|---|---|---|---|---|---|
| **LCM** | ✗ (Proposition 1) | ✗ (Proposition 4) | $n = 2$: $(2 - \sqrt{2})^2 \approx 0.3431$ (Theorem 2) | $n \geq 3$: $\left( (1 - \frac{1}{n}^{\frac{1}{n-1}})^n,\ (1 - \gamma^2)^n \right]^*$ (Theorem 3) | 0 (Theorem 4) | |
| **PVM** | ✓ (Proposition 2) | ✓ (Proposition 3) | $n = 2$: $3/4 = 0.75$ (Theorem 6) | $n \geq 3$: $1 - \left(1 - \frac{1}{n}\right)\left(\frac{1}{n}\right)^{\frac{1}{n-1}}$ (Theorem 6) | $n = 2$: $19/27 \approx 0.7037$ (Theorem 7) | $n \geq 3$: $\left[ (\frac{19}{27})^{\lceil \log_2 n \rceil},\ 1 - \left(1 - \frac{1}{n}\right)\left(\frac{1}{n}\right)^{\frac{1}{n-1}} \right]$ (Theorem 6 & 8) |

$^*\gamma = \sqrt[3]{\frac{2+\sqrt{6}}{4}} + \sqrt[3]{\frac{2-\sqrt{6}}{4}}$).

We begin by analyzing the commonly used Last-Click Mechanism (LCM) and theoretically demonstrate that it is not DSIC. For LCM's performance, our findings reveal that, in the worst-case scenario,

[2]This shift has been driven by privacy regulations such as the European Union's *General Data Protection Regulation* and Apple's *App Tracking Transparency* framework [8, 2, 16], and the adoption of redirect-less systems including Google's and Microsoft's *parallel tracking* and Apple's *SKAdNetwork* [10, 19, 3].

LCM can perform remarkably poorly. Even with just two heterogeneous platforms, both accuracy and fairness can approach arbitrarily low values.

To ensure DSIC, we propose a novel attribution mechanism called the Peer-Validated Mechanism (PVM). The mechanism operates as follows: only platforms reporting before the conversion are eligible for attribution, and the credit a platform receives depends solely on peer reports and prior probabilities—independent of its own report. Since a platform's report does not influence its own outcome, PVM is DSIC by design. We then theoretically demonstrate that PVM consistently outperforms the LCM in terms of both attribution accuracy and fairness. We further prove that it is the optimal DSIC mechanism in the homogeneous setting (Theorem 5). Mutiple simulations using distributions fitted from real-world ad-conversion data further validate the superiority of PVM.

To the best of our knowledge, this is the first work to formally model the advertising attribution problem within a theoretical framework, and to rigorously analyze the incentive and efficiency properties of the widely adopted Last-Click Mechanism. By shifting attention from empirical heuristics and estimation to mechanism design, our work offers foundational insights for developing attribution systems that are robust, fair, and incentive-compatible in digital advertising markets.

All missing proofs can be found in full version.

**Related Work**   Recent research on advertising attribution has primarily focused on multi-touch attribution, which distributes conversion credit across multiple platforms based on observed user interactions data. A wide range of modeling approaches have been explored, including probabilistic models such as survival analysis [13, 14, 23, 28, 29], Shapley value-based methods for fair allocation [1, 4, 24], and Markov models for channel transition influence [1, 15]. Furthermore, causal inference [6, 27] and deep learning [21, 18, 6, 17, 26, 27] have been applied to better capture temporal and interaction complexity. Despite their sophistication, these approaches generally assume that the user interactions data reported by platforms are accurate and complete.

However, this assumption often fails in practice, as platforms may strategically misreport to gain greater attribution. In contrast, mechanism design offers a principled framework for addressing strategic behavior, with incentive compatibility (IC) as a central design goal [12, 20, 25, 5, 11]. While IC-based techniques have been widely applied in domains such as auctions [7], voting [9], and resource allocation [22], their application to attribution remains underexplored. Attribution presents new challenges: the strategic behavior of platforms is often ill-defined, and their utility depends on uncertain conversion outcomes, making standard mechanism design tools difficult to apply directly.

## 2   Model and Preliminaries

This section develops a formal model to study advertising attribution under strategic platform behavior. We first describe a typical real-world scenario, then formalize the model components, define the attribution mechanism, analyze strategic behavior, and finally define the advertiser's objective.

Throughout, we adopt the last-click attribution standard, treating the final platform in a user's interaction sequence as the one that deserves credit. We focus on settings with at least two platforms, the minimal case where attribution ambiguity and manipulation arise. In practice, the number of platforms involved in a conversion is typically small—often no more than five.

### 2.1   Real-World Advertising Scenario

Consider the real-world online advertising scenario where a user interacts with ads from multiple platforms ($n \geq 2$) before a conversion event. When the user converts, the advertiser seeks to allocate credit based on the click logs reported by the platforms.

The process unfolds as follows. Each platform $i \in [n]$ first detects a user click and records a log at the corresponding *absolute click time* $t_i^{\mathrm{abs}}$. It then selects an *absolute reported time* $r_i^{\mathrm{abs}} \geq t_i^{\mathrm{abs}}$, at which it submits the log to the advertiser. At some time $t_0 \geq \max_{i \in [n]}\{t_i^{\mathrm{abs}}\}$, the user converts, and the advertiser performs credit attribution based on reports received by $t_0$—that is the set $\{r_i^{\mathrm{abs}} \mid r_i^{\mathrm{abs}} \leq t_0\}$. Crucially, while the advertiser observes the conversion time $t_0$, platforms must decide when to report without knowing when the conversion will occur.

This scenario highlights the fundamental challenge in attribution: the advertiser must infer the true sequence of events based on potentially delayed reports from strategically acting platforms. The discrepancy between true click times and reported times necessitates careful mechanism design.

## 2.2 Advertising Attribution Model

We now present a game-theoretic model of the attribution process, capturing strategic platform behavior and informing mechanism design. To simplify analysis, we adopt a *conversion-aligned timeline*, setting the conversion time $t_0 = 0$ without loss of generality. Under this transformation, all click times are expressed relative to the conversion and lie in $(-\infty, 0]$. Specifically, we define

$$t_i := t_i^{\text{abs}} - t_0 \leq 0, \quad \forall i \in [n],$$

where $t_i$ denotes the relative click time of platform $i$. Figure 1 illustrates this transformation: if two platforms record clicks at 10:40 a.m. and 10:50 a.m., and the conversion occurs at 11:00 a.m., their relative click times become $t_1 = -20$ and $t_2 = -10$, with the conversion at time 0. Under the

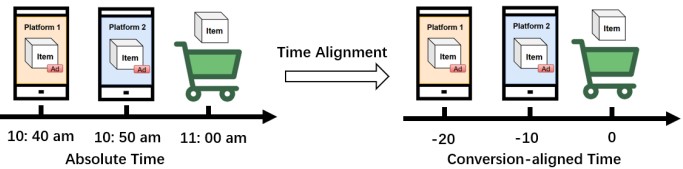

Figure 1: Conversion-aligned time transformation

conversion-aligned timeline, the relative click time $\boldsymbol{t} = (t_i)_{i \in [n]}$—which depends on the unknown conversion time—is therefore unobservable to platforms. To capture this uncertainty, we model $t_i$ as a random variable drawn independently from a commonly known distribution, with cumulative distribution function (CDF) $F_i(t)$ and probability density function (PDF) $f_i(t)$, supported on $(-\infty, 0]$. This distribution may be interpreted as a prior belief based on platform-level statistics. Let $\boldsymbol{F} = \{F_i\}_{i \in [n]}$ denote the joint distribution from which the click time vector $\boldsymbol{t}$ is drawn.

To model strategic reporting, we assume each platform $i \in [n]$ selects a non-negative *reporting delay* $\tau_i \geq 0$, resulting in a reported time $r_i = t_i + \tau_i$ on the conversion-aligned timeline[3]. Since $t_i$ is unobservable when the platform commits to its strategy, the chosen delay $\tau_i$ is applied uniformly across all realizations of $t_i$. We denote the delay profile as $\boldsymbol{\tau} = (\tau_i)_{i \in [n]}$, and the resulting reported time profile by $\boldsymbol{r} = (r_i)_{i \in [n]} = \boldsymbol{t} + \boldsymbol{\tau}$. Unless stated otherwise, all subsequent analysis is conducted on the conversion-aligned timeline.

## 2.3 Attribution Mechanism

We define an attribution mechanism $\mathcal{M}$ by its assignment rule $\mathcal{M}(\boldsymbol{r}) := \{x_i(\boldsymbol{r})\}_{i \in [n]}$, where $x_i(\boldsymbol{r})$ denotes the credit assigned to platform $i$ given the reported time profile $\boldsymbol{r}$.[4] A mechanism is said to be *feasible* if it satisfies the following constraints:

$$0 \leq x_i(\boldsymbol{r}) \leq 1, \quad \forall i \in [n], \boldsymbol{r} \tag{1}$$

$$x_i(r_i, \boldsymbol{r}_{-i}) = 0, \quad \forall i \in [n], r_i > 0, \boldsymbol{r}_{-i} \tag{2}$$

$$\mathbb{E}_{\boldsymbol{t} \sim \boldsymbol{F}} \left[ \sum_{i=1}^{n} x_i(\boldsymbol{t} + \boldsymbol{\tau}) \right] \leq 1, \quad \forall \boldsymbol{F}, \boldsymbol{\tau} \tag{3}$$

Constraints (1) and (2) bound individual credit and exclude post-conversion reports. Constraint (3) limits the expected credit, ensuring that the advertiser's overall budget is respected.[5]

---

[3] We focus on strategic *delay* in reporting rather than repeated or fraudulent submissions. Each platform reports its click once, possibly after a strategic delay, to maximize its attribution credit. This setting differs from *click spamming* and *click injection*, which involve multiple or fabricated reports.

[4] In practice, reports submitted after the conversion time (i.e., $r_i > 0$) are typically not received or used by the advertiser. However, for modeling generality, we allow such values as inputs to the mechanism. Their exclusion from attribution is later enforced through explicit feasibility constraints (see Constraint (2)).

[5] Constraint (3) normalizes total credit in expectation rather than per conversion. This design reflects advertisers' long-term budget control: it maintains the average expenditure over many conversions while allowing more flexibility for designing incentive-compatible mechanisms than strict per-instance normalization.

Given an attribution mechanism $\mathcal{M}$, we define platform $i \in [n]$'s instantaneous utility under report profile $\boldsymbol{r}$ as the credit assigned to it by the mechanism: $u_i(\boldsymbol{r}) = x_i(\boldsymbol{r})$. Given the distribution profile $\boldsymbol{F}$ and others' strategy $\boldsymbol{\tau}_{-i}$, platform $i$ selects its delay $\tau_i$ to maximize its expected utility:

$$U_i(\tau_i, \boldsymbol{\tau}_{-i}) = \mathbb{E}_{\boldsymbol{t} \sim \boldsymbol{F}}\left[u_i(\boldsymbol{t} + \boldsymbol{\tau})\right] = \mathbb{E}_{\boldsymbol{t} \sim \boldsymbol{F}}\left[x_i(\boldsymbol{t} + \boldsymbol{\tau})\right].^6$$

From the advertiser's perspective, an ideal attribution mechanism $\mathcal{M}$ should achieve two primary goals: (i) incentivize truthful reporting to ensure reliable interaction data; and (ii) accurately assign credit to the platform responsible for the conversion.

We first use *dominant strategy incentive compatibility* (DSIC) to capture truthful reporting:

**Definition 1** (DSIC). *A mechanism $\mathcal{M}$ is DSIC if for every platform $i$, truthful reporting ($\tau_i = 0$) maximizes its utility $u_i(\boldsymbol{r})$ regardless of the realized true click times $\boldsymbol{t} = (t_i, \boldsymbol{t}_{-i})$ or the strategies $\boldsymbol{\tau}_{-i}$ chosen by other platforms[7]. That is, for all platforms $i \in [n]$, all true times $t_i \leq 0$ and $\boldsymbol{t}_{-i}$, all others' strategies $\boldsymbol{\tau}_{-i}$, and any deviation delay $\tau_i' > 0$:*

$$u_i(t_i, \boldsymbol{t}_{-i} + \boldsymbol{\tau}_{-i}) \geq u_i(t_i + \tau_i', \boldsymbol{t}_{-i} + \boldsymbol{\tau}_{-i}).$$

It is easy to verify that DSIC is equivalent to a non-increasing allocation rule with respect to a platform's own report.

**Theorem 1.** *An attribution mechanism $\mathcal{M}$ satisfies DSIC if and only if, for every platform $i$ and any fixed reports from other platforms $\boldsymbol{r}_{-i}$, the credit $x_i(r_i, \boldsymbol{r}_{-i})$ is non-increasing in its own report $r_i$.*

Second, we formalize attribution accuracy as the mechanism's ability to assign credit to the true last-click platform. Specifically, we defined the accuracy of a mechanism $\mathcal{M}$, given $\boldsymbol{F}$ as

$$\text{ACC}(\mathcal{M}; \boldsymbol{F}) = \mathbb{E}_{\boldsymbol{t} \sim \boldsymbol{F}}\left[\sum_{i=1}^{n} x_i(\boldsymbol{t} + \boldsymbol{\tau}^{\text{NE}}) \cdot \mathbb{I}[i = \arg\max_j\{t_j\}]\right],$$

where $\boldsymbol{\tau}^{\text{NE}}$ is the Nash equilibrium induced by $\boldsymbol{F}$ and $\mathcal{M}$. Note that $\boldsymbol{\tau}^{\text{NE}} = \boldsymbol{0}$ for a DSIC mechanism. Thus, given a known distribution $\boldsymbol{F}$, the advertiser's optimization problem is formulated as:

$$\begin{aligned} \max_{\mathcal{M}} \quad & ACC(\mathcal{M}; \boldsymbol{F}) \\ \text{s.t.} \quad & \text{Feasible and DSIC.} \end{aligned} \tag{4}$$

Beyond defining accuracy with respect to a fixed $\boldsymbol{F}$, we define the accuracy of mechanism $\mathcal{M}$ as

$$ACC(\mathcal{M}) := \inf_{\boldsymbol{F}} ACC(\mathcal{M}; \boldsymbol{F}),$$

capturing worst-case performance across $\boldsymbol{F}$, serving as a evaluation for a mechanism's performance.

## 3 The Last-Click Mechanism

In this section, we conduct a rigorous analysis of the Last-Click Mechanism (LCM). We begin by formally defining LCM and then demonstrate that it fails to satisfy DSIC. We further evaluate its accuracy at equilibrium and derive accuracy bounds in both homogeneous and heterogeneous platform settings. Formally, the Last-Click Mechanism is defined as $\mathcal{M}_{\text{LCM}}$:

**Definition 2** (Last-Click Mechanism). *Given the report profile $\boldsymbol{r} = (r_i)_{i \in [n]}$, the Last-Click Mechanism is defined as $\mathcal{M}_{LCM} = \{x_i(\boldsymbol{r})\}_{i \in [n]}$. Specifically,*

$$x_i(\boldsymbol{r}) = \begin{cases} 1 & \text{if } i \in S \text{ and } r_i = \max_{j \in S}\{r_j\}, \\ 0 & \text{otherwise,} \end{cases}$$

*where $S = \{j \in [n] \mid r_j \leq 0\}$ is the set of platforms with effective reports. Ties are broken uniformly at random among the tied platforms.*

---

[6]Note that, in the absolute-time model, the platform must take expectation over the unknown conversion time and other click times. In the conversion-aligned model, the conversion time is fixed at 0, and the same uncertainty is reflected in the distribution of $\boldsymbol{t}$.

[7]DSIC benefits cold-start scenarios by ensuring truthful reporting without requiring prior knowledge, enabling reliable attribution from the outset and facilitating the learning of the true distribution.

Due to its simplicity and its intuitive principle of crediting the platform associated with the user's final click before conversion, LCM is widely adopted in practice. However, it is easy to see that platforms may benefit from strategically delaying their reports, making truthful reporting suboptimal. Therefore, LCM does not satisfy DSIC.

**Proposition 1.** *The Last-Click Mechanism is not a DSIC mechanism.*

### 3.1 Accuracy Analysis

Since LCM is not DSIC, it may assign credit to a platform that wasn't truly last, leading to inaccurate attribution. We therefore analyze its equilibrium accuracy in both homogeneous and heterogeneous platform settings, and derive accuracy bounds for both two-platform and $n$-platform cases.[8]

We first consider the case with two homogeneous platforms and present our result in Theorem 2.

**Theorem 2.** *When there are two homogeneous platforms with identical distribution $F(t)$, supported on $(-\infty, 0]$, the accuracy of $\mathcal{M}_{LCM}$ is exactly $(2 - \sqrt{2})^2$, and this bound is tight.*

To prove Theorem 2, we first analyze the incentive constraint at a symmetric strategy profile $(\tau_0, \tau_0)$. By requiring that no platform benefits by unilaterally deviating from $\tau_0$ to truthful reporting, we derive the necessary condition $F(-\tau_0) \geq 2 - \sqrt{2}$. Since LCM can only attribute correctly when both true click times are before $-\tau_0$, this yield a lower bound on accuracy of $(2 - \sqrt{2})^2$. We then construct a family of distributions $f_M(t) = c_M(e^{-t} - 1)$, supported on $[-M, 0]$, where $c_M$ is a normalization constant. We show that this game admits a unique symmetric Nash equilibrium, and as $M \to \infty$, the accuracy converges to exactly $(2 - \sqrt{2})^2$.

We now extend our analysis to the general case with $n$ homogeneous platforms.

**Theorem 3.** *When there are $n$ homogeneous platforms with identical distribution $F(t)$, supported on $(-\infty, 0]$, the accuracy of $\mathcal{M}_{LCM}$ is bounded as follows:*

$$\left(1 - \left(\frac{1}{n}\right)^{\frac{1}{n-1}}\right)^n < ACC(\mathcal{M}_{LCM}) \leq \left(1 - \left(\sqrt[3]{\frac{2 + \sqrt{6}}{4}} + \sqrt[3]{\frac{2 - \sqrt{6}}{4}}\right)^2\right)^n.$$

The lower bound is derived using an argument similar to the two-platform case, by examining the conditions required for a symmetric equilibrium. For the upper bound, we analyze the symmetric equilibrium under a specific distribution with a linear probability density function $f(t) = -2t$ supported on $[-1, 0]$.

Finally, we consider the heterogeneous case, where each platform may follow a different distribution $F_i(t)$. Surprisingly, we show that the accuracy of the LCM can be arbitrarily low, even in a simple setting with just two heterogeneous platforms.

**Theorem 4.** *When there are $n$ heterogeneous platforms with distributions $F_i(t)$, all supported on $(-\infty, 0]$. the accuracy of $\mathcal{M}_{LCM}$ can be arbitrarily small and approach to 0.*

The proof relies on a key insight: a platform with a highly concentrated distribution (e.g., supported on $(C - \epsilon, C + \epsilon)$) can easily manipulate its report to secure attribution credit. We construct an instance where one platform has such a concentrated distribution, while the others have click time supports strictly greater than it. In this scenario, we show that in equilibrium, the concentrated platform receives attribution with probability approaching 1, causing overall accuracy to approach 0.

## 4 The Peer-Validated Mechanism

In this section, we introduce the Peer-Validated Mechanism (PVM), a novel mechanism addressing the non-DSIC issue of LCM. Intuitively, if the credit assigned to a platform is independent of its own report, the mechanism is DSIC. Based on this idea, we propose the PVM as follows:

---

[8]Since our focus is on DSIC mechanisms, we restrict our evaluation of LCM to instances where equilibrium is guaranteed, without analyzing its existence in general. Even within this limited scope, the results clearly demonstrate LCM's poor performance in our setting.

**Definition 3** (Peer-Validated Mechanism). *Consider $n$ platforms with the CDF $\{F_i\}_{i \in n}$ and PDF $\{f_i\}_{i \in n}$ supported on $(-\infty, 0]$. Let $\boldsymbol{r} = (r_i)_{i \in [n]}$ be the reported time profile from $n$ platforms. The Peer-Validated Mechanism assigns credit based on mutual validation among platforms, and is defined as $\mathcal{M}_{PVM} = \{x_i(\boldsymbol{r})\}_{i \in [n]}$, with*

$$x_i(\boldsymbol{r}) = \begin{cases} \mathbb{I}[r_i \leq 0] \cdot \mathbb{I}[\max_{j \in S \setminus \{i\}}\{r_j\} \leq \alpha_S^{(i)}] & \text{if } |S \setminus \{i\}| \geq 1, \\ \mathbb{I}[r_i \leq 0] \cdot \beta_i & \text{otherwise,} \end{cases}$$

*where $S = \{j \in [n] \,|\, r_j \leq 0\}$ denotes platforms with eligible reports. $\beta_i = P(i = \arg\max_j\{t_j\}) = \int_{-\infty}^{0} f_i(t) \prod_{j \neq i} F_j(t) \, dt$ is the probability that platform $i$ is the true last-click platform based on the prior. The validation threshold $\alpha_S^{(i)}$ is defined as the solution to $\prod_{j \in S \setminus \{i\}} F_j(\alpha_S^{(i)}) = \beta_i$.[9]*

Roughly speaking, PVM assigns credit to platform $i$'s credit based on eligible reports from other platforms. When such peer reports exist, the mechanism compares them to a threshold $\alpha_S^{(i)}$, which is chosen so that the probability of all peers' true click times being no later than $\alpha_S^{(i)}$ matches the prior $\beta_i$ that platform $i$ is the true last. This validation process leverages instance-level information to make attribution decisions while preserving incentive compatibility. If no eligible peer reports are available, PVM falls back to allocating $\beta_i$ based on the prior. The reason only eligible reports are used for validation is that the mechanism assumes no overt misreporting among them, while ineligible reports ($r_j > 0$) are definitely misreports and thus excluded due to unmodeled behavior. Finally, the indicator $\mathbb{I}[r_i \leq 0]$ ensures that attribution only goes to pre-conversion reports.

Since any reporting delay either disqualifies the platform itself or prevents others from being attributed, it is straightforward to verify that PVM satisfies feasibility and DSIC [10] , as formalized in Proposition 2.

**Proposition 2.** *The Peer-Validated Mechanism is a DSIC mechanism.*

As PVM is DSIC, we focus on truthful reports ($\boldsymbol{r} = \boldsymbol{t}$). In this case, $S = [n]$, and we let $\alpha_i$ denote the threshold used in $x_i(\cdot)$, defined by $\prod_{j \neq i} F_j(\alpha_i) = \beta_i$. The allocation rule then simplifies to

$$x_i(\boldsymbol{t}) = \mathbb{I}[\max_{j \neq i}\{t_j\} \leq \alpha_i] \quad \forall i \in [n].$$

We adopt this reduced form throughout the remainder of our analysis of PVM.

### 4.1 Optimality of PVM for Homogeneous Platforms

We surprisingly find that PVM is the optimal DSIC mechanism in the homogeneous platform setting.

**Theorem 5.** *When the platforms are homogeneous, the Peer-Validated Mechanism (PVM) is the optimal DSIC mechanism with respect to the accuracy.*

To show this optimality, we aim to identify the DSIC attribution rule $\{x_i(\boldsymbol{t})\}_{i \in [n]}$ that maximizes accuracy. This is a challenging task, as it involves optimizing over a set of functions simultaneously. However, if for any fixed expected attribution $e_i = \mathbb{E}_{\boldsymbol{t}}[x_i(\boldsymbol{t})]$, we can characterize the most accurate DSIC rule that achieves it, then the problem reduces to optimizing over the expected attribution vector $\boldsymbol{e} = (e_i)_{i \in [n]}$. The following lemma shows that such a characterization indeed exists.

**Lemma 1.** *For platform $i$ and a fixed expected attribution $e_i$, there exists an optimal DSIC attribution rule for platform $i$ w.r.t. accuracy, satisfying $e_i = \mathbb{E}_{\boldsymbol{t}}[x_i(\boldsymbol{t})]$, that can be written as*

$$x_i^*(t_i, \boldsymbol{t}_{-i}) = \begin{cases} 1, & \text{if } \max_{j \neq i}\{t_j\} \leq \theta_i, \\ 0, & \text{otherwise,} \end{cases}$$

*where $G_i(t) = \Pi_{j \neq i} F_j(t)$ is the CDF of the random variable $\max_{j \neq i}\{t_j\}$, and $G_i(\theta_i) = e_i$.*

---

[9]The existence and uniqueness of $\alpha_S^{(i)}$ and $\beta_i$ under standard regularity conditions, along with handling edge cases (e.g., flat or discontinuous CDFs), are detailed in full version.

[10]A simple variant of PVM also preserves DSIC under the click spamming problem, where a platform may repeatedly report the same click at later timestamps. Specifically, the modified mechanism takes the first valid report (if any) from each platform as input while keeping the rest of the allocation rule unchanged. Under this setting, the platform's own reporting time remains decoupled from its expected number of attributions.

Herein, we give a proof sketch of Lemma 1. First, as any DSIC rule must be non-increasing in $t_i$ (Theorem 1), we claim that, to maximize accuracy, the optimal DSIC rule $x_i(t_i, \boldsymbol{t}_{-i})$ should be a constant when given $\boldsymbol{t}_{-i}$, so that larger values of $t_i$, which more better indicate that platform $i$ is last, are not penalized. Second, given a fixed expected attribution $e_i$, the self-independent rule $x_i(\boldsymbol{t}_{-i})$ should prioritize instances with smaller $\max_{j \neq i}\{t_j\}$, where platform $i$ is more likely to be last. This greedy strategy yields the threshold-form optimal DSIC rule in the lemma. When $G_i$ is somewhere flat, multiple thresholds may achieve $e_i$, and combining them may yield non-threshold variants. Still, at least one such optimal rule exists.

Based on Lemma 1, the task reduces to finding the optimal $(e_i^*)_{i \in [n]}$. Since $G_i(\theta_i) = e_i$, the original optimization problem (4) can therefore be reformulated in terms of $\boldsymbol{\theta} = (\theta_i)_{i \in [n]}$ as follows:

$$\max_{\boldsymbol{\theta}} \quad \sum_{i=1}^{n} \int_{-\infty}^{\theta_i} g_i(u)(1 - F_i(u))\, du$$

$$\text{s.t.} \quad \sum_{i=1}^{n} G_i(\theta_i) = 1$$

In particular, for homogeneous platforms, the thresholds defined within the PVM precisely align with the solution to the optimization problem outlined above, establishing its optimality as stated in Theorem 5.

## 4.2 Accuracy Analysis

We now analyze the accuracy of PVM. For the homogeneous setting, we give a tight bound on the accuracy. Since PVM is the optimal DSIC mechanism, this accuracy is the maximum value a DSIC mechanism can achieve.

**Theorem 6.** *When there are $n$ homogeneous platforms with identical distribution $F(t)$, supported on $(-\infty, 0]$, the accuracy of $\mathcal{M}_{PVM}$ is exactly equal to*

$$ACC(\mathcal{M}_{PVM}) = 1 - \left(1 - \frac{1}{n}\right)\left(\frac{1}{n}\right)^{\frac{1}{n-1}}.$$

In the homogeneous case, symmetry implies that all thresholds $\alpha_i$ are equal, denoted by $\alpha$, and satisfy $F(\alpha)^{n-1} = 1/n$. Therefore, PVM makes a correct attribution if either all reports are no greater than $\alpha$, which occurs with probability $(1/n)^{n/(n-1)}$, or exactly one report exceeds $\alpha$, which occurs with probability $1 - (1/n)^{1/(n-1)}$. These probabilities depend only on $n$, not on the specific distribution. Summing them gives the accuracy in Theorem 6.

In practice, the number of platforms $n$ typically does not exceed 5. We therefore conduct a comparison with the Last-Click mechanism (presented in Table 2) to show that PVM is strictly superior.

Table 2: The accuracy comparison between PVM and LCM (Upper bound).

| $n$ | $\mathcal{M}_{\text{PVM}}$ | Upper bound of $\mathcal{M}_{\text{LCM}}$ | Ratio ($\mathcal{M}_{\text{PVM}}$ / $\mathcal{M}_{\text{LCM}}$ UB) |
|---|---|---|---|
| 2 | 0.75 | $(2 - \sqrt{2})^2 \approx 0.3431$ (tight bound) | 2.1857 |
| 3 | 0.6151 | 0.3336 | 1.8437 |
| 4 | 0.5275 | 0.2314 | 2.2799 |
| 5 | 0.4650 | 0.1605 | 2.8977 |

In the rest, we consider the general heterogeneous-platform setting. We show a tight bound on accuracy for two-platform case (Theorem 7) and a lower bound for $n$-platform case (Theorem 8).

**Theorem 7.** *When there are two heterogeneous platforms, the accuracy of $\mathcal{M}_{PVM}$ is exactly equal to $ACC(\mathcal{M}_{PVM}) = 19/27 \approx 0.7037$.*

To establish the result, we first formulate an optimization problem that characterizes the worst-case accuracy by maximizing the misattribution probability. In the two-platform setting, all attribution outcomes can be explicitly enumerated, making this optimization analytically tractable. To show tightness, we then construct a concrete instance that satisfies the optimality conditions, thereby achieving the accuracy value of $19/27$.

**Theorem 8.** *When there are $n$ heterogeneous platforms, the lower bound on the accuracy of $\mathcal{M}_{PVM}$ is $ACC(\mathcal{M}_{PVM}) = (19/27)^{\lceil \log_2 n \rceil}$.*

For $n$ heterogeneous platforms, we design a binary-tree-based mechanism to derive a lower bound for PVM. Starting from the root node, which represents all $n$ platforms, we recursively partition them into two disjoint subsets $L$ and $R$ of sizes $\lceil n/2 \rceil$ and $\lfloor n/2 \rfloor$, respectively. Each subset is treated as a virtual platform, represented by the distribution of $\max_{i \in L}\{t_i\}$ or $\max_{i \in R}\{t_i\}$. At each internal node, the attribution reduces to a problem between two heterogeneous platforms. Repeating this over $\lceil \log_2 n \rceil$ levels yields an overall accuracy lower bound of $(19/27)^{\lceil \log_2 n \rceil}$. Since PVM is guaranteed to perform at least as well as this mechanism, the same expression serves as a lower bound for its accuracy.

### 4.3 Fairness of PVM

Besides the DSIC and accuracy, PVM also satisfies a strong *fairness* property: the expected attribution $\mathbb{E}[x_i]$ for each platform $i$ exactly matches its true probability of contributing the last click, $P(i = \arg\max_j\{t_j\})$. This alignment offers a principled basis for evaluating long-term platform effectiveness and simultaneously promotes trust in the mechanism's equity. To quantify this alignment and enable comparisons across mechanisms, we define the following metric:

**Definition 4.** *The* fairness score *of mechanism $\mathcal{M}$ under the joint distribution $\boldsymbol{F}$ is defined as*

$$FAIR(\mathcal{M}; \boldsymbol{F}) = \min_{\{i \mid P(i=\arg\max_j\{t_j\})>0\}} \left\{ \frac{\mathbb{E}[x_i(\boldsymbol{t})]}{P(i = \arg\max_j\{t_j\})} \right\}.$$

**Definition 5.** *A mechanism $\mathcal{M}$ is* Fair *if, for any joint distribution $\boldsymbol{F}$, it holds that*

$$FAIR(\mathcal{M}; \boldsymbol{F}) = 1.$$

The fairness score $FAIR(\mathcal{M}; \boldsymbol{F})$ quantifies how closely a mechanism's expected attribution matches the true last-click probabilities, with a score of 1 indicates perfect alignment. A *Fair* mechanism ensures that attribution faithfully reflects contribution probabilities across all distributions. PVM is a fair mechanism directly from the choice of $(\alpha_i)_{i \in [n]}$ under DSIC:

$$\mathbb{E}_{\boldsymbol{t}}[x_i(\boldsymbol{t})] = \mathbb{E}_{\boldsymbol{t}}[\max_{j \neq i}\{t_j\} \leq \alpha_i] = \prod_{j \neq i} F_j(\alpha_i) = P(i = \arg\max_j\{t_j\}).$$

**Proposition 3.** *The Peer-Validated Mechanism is* Fair.

In contrast, the Last-Click Mechanism fails to meet this property.

**Proposition 4.** *The Last-Click Mechanism is not* Fair.

LCM fails the *Fair* property due to its fairness score being highly sensitive to distributional differences and strategic delays, especially under heterogeneity. As shown in Table 3, the fairness score can degrade to zero in such settings.

Table 3: Worst-Case Fairness Score of LCM under Equilibrium.

| Scenario | Worst-Case Fairness ($\inf_{\boldsymbol{F}} FAIR(\mathcal{M}_{LCM}; \boldsymbol{F})$) |
|---|---|
| Homogeneous, $n = 2$ | $1 - (\sqrt{2}-1)^2 \approx 0.828$ |
| Homogeneous, $n \geq 3$ | $(1 - (1/n)^{n/(n-1)}, 1 - (\sqrt[3]{\frac{2+\sqrt{6}}{4}} + \sqrt[3]{\frac{2-\sqrt{6}}{4}})^{2n}]$ |
| Heterogeneous, $n \geq 2$ | $0$ |

## 5 Numerical Experiments

We empirically evaluate PVM against LCM using simulations based on click time distributions fitted from real-world ad conversion logs from four advertising platforms. Experiments cover two settings: homogeneous and heterogeneous. In the homogeneous case, we simulate $n \in \{2, 3, 4, 5\}$ identical platforms, all following the click time same distribution, repeated across four distributions derived

from real data. In the heterogeneous case with $n = 2$, we simulate all six platform pairs formed by different combinations of the four distributions. Under LCM, platforms play in equilibrium; under PVM, they report truthfully by DSIC. Each configuration was evaluated using $5 \times 10^4$ simulated user paths, repeated over 10 independent runs.

PVM consistently outperforms LCM in both accuracy and fairness across all settings. Table 4 reports the improvements as mean $\pm$ standard deviation over platforms (homogeneous) or platform pairs (heterogeneous). Specifically, accuracy gains grew with $n$ (up to $0.3041$ when $n = 5$) and remains notable under heterogeneity ($0.0655$). Fairness improvements are small in homogeneous cases but substantial in heterogeneous ones ($0.1320$).

Table 4: Aggregate summary of PVM's improvements over LCM, (mean $\pm$ standard deviation)

| Metric | Homo Setting | | | | Hetero Setting |
| | $n = 2$ | $n = 3$ | $n = 4$ | $n = 5$ | (over 6 pairs) |
| --- | --- | --- | --- | --- | --- |
| Acc. | $0.0404 \pm 0.0396$ | $0.1583 \pm 0.0439$ | $0.2444 \pm 0.0580$ | $0.3041 \pm 0.0490$ | $0.0655 \pm 0.0283$ |
| Fair. | $0.0248 \pm 0.0089$ | $0.0157 \pm 0.0034$ | $0.0107 \pm 0.0047$ | $0.0111 \pm 0.0040$ | $0.1320 \pm 0.0598$ |

## 6 Conclusion and Discussion

This paper introduces a formal game-theoretic framework for advertising attribution under strategic platform behavior. We show that the widely used Last-Click Mechanism fails to be dominant strategy incentive-compatible (DSIC) and performs poorly in both accuracy and fairness. To address these limitations, we propose the Peer-Validated Mechanism (PVM), a novel DSIC mechanism that allocates credit based on peer reports. We prove that PVM achieves optimal accuracy in homogeneous settings, offers provable guarantees in heterogeneous ones, and satisfies a strong fairness property. Our theoretical analysis is further validated by numerical experiments using real-world data, where PVM consistently outperforms LCM.

In practice, peer-validation principle offers a concrete design guideline for incentive-compatible attribution systems. For instance, in machine learning-based models, excluding a platform's own report as an input feature ensures truthfulness, shifting the focus from detection to design.

Besides, PVM framework can be extended to settings with correlated click-time distributions while preserving the core peer-validation principle and the DSIC property. The validation rule generalizes from a scalar threshold ($\alpha_i$) to a multi-dimensional acceptance region $D_i$ over peer reports $t_{-i}$, constructed greedily by including outcomes with the highest posterior probability that platform $i$ was the true last click until $P(t_{-i} \in D_i) = \beta_i$. A platform receives credit if and only if its peers' reports fall within $D_i$. Under this modification, the homogeneous-case results remain unchanged, since our proofs for those theorems do not rely on the independence assumption; the results for the last-click mechanism also remain the same, as its accuracy and fairness are already zero; while in heterogeneous settings, PVM retains a weaker but still meaningful $1/n$ lower bound on accuracy. We focus on the independence assumption in this paper to present the mechanism's core insight in the clearest setting, which is sufficient to capture the essential strategic structure, leaving correlated extensions for future work.

Several directions remain open. First, while PVM aligns the expected attribution with true last-click probability, future work may explore mechanisms that further improve instance-level accuracy. Second, investigating correlated click-time distributions could enhance a mechanism's applicability in realistic scenarios. Next, a joint optimization framework modeling both advertiser and platform utilities, integrating attribution with budget allocation, represents a compelling direction. Finally, investigating repeated games with externalities—where platforms may strategically harm peers or misreport distributions to manipulate learned priors—could address dynamic interactions, potentially incorporating bidding strategies for a more comprehensive ecosystem model.

## Acknowledgment

This work was supported by National Natural Science Foundation of China (No.62472428), Public Computing Cloud, Renmin University of China, the fund for building world-class universities (disciplines) of Renmin University of China.

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
