# OpenReview forum: "Beyond Last-Click: An Optimal Mechanism for Ad Attribution"
_NeurIPS.cc/2025/Conference — NeurIPS 2025 poster_

### Official Review · Reviewer_Tb4U · 2025-06-30

**Clarity:** 3
**Significance:** 3
**Originality:** 3
**Rating:** 4
**Confidence:** 4

**Summary:**

The paper provides a mechanism design perspective on the last click attribution mechanism in online advertising.
It frames the situation of click reporting as a mechanism design problem where platforms have incentive to misreport their times of interaction with the user in order to get the attribution.
A prior on the times of interaction is assumed (Bayesian model). The Accuracy of the mechanism is defined as a performance metrics.
After showing that the Last Click mechanism is not incentive compatible, the paper introduces an alternative, PVM (peer-validated mechanism) that satisfies this property. The paper then consider the homogeneous case (iid) to refine the results on accuracy.
The paper finishes with some consideration on fairness and a very short simulation section.

**Questions:**

# Major question

Can you please provide more details on where the situation described in the paper occurs in practice?
Maybe   providing examples of what you call a platform would help me better contextualize.


# Minor questions

Do you have real-life example of such strategy being applied somewhere (l37)

In practice, what is the mechanism by which a platform declares a click, and what  can the advertiser  checks on the platform declaration ?

Can the mechanism be adapted in the case where multiple conversions or clicks per platform occur?

**Ethical Concerns:**

["NO or VERY MINOR ethics concerns only"]

**Final Justification:**

The authors addressed my major concern on the paper relevancy on real life situation.
I am therefore increasing my score (by 2).

**Paper Formatting Concerns:**

nope

**Quality:**

2

**Strengths And Weaknesses:**

The paper is quite original as it presents a mechanism design problem where the hidden information is not related to the agents' preferences. The topic of last-click attribution is indeed an important one, as it is a heuristic known for having many failure modes. For instance, an RTB bidder is incentivised to overbid when it knows it lost the attribution. The introduction of measures of fairness and accuracy seems novel in this context. The peer validated mechanism the authors identified is simple and optimal. Also, it seems that  since it is non-increasing, it is DSIC, which implies that even if the advertiser do not have the right priors at cold-start, it will be able to learn them from the generated data.


I have some strong reservation on this paper, because the situation that motivates the analysis is  not  how things work in practice. I hope I missed something, in which case a clarification from the authors should change my evaluation.
The blocking issue to me is the following sentence, that motivates most of the document: "In particular, a platform may delay the reporting of a user’s click to appear later in the user’s interaction sequence and thereby capture the conversion credit."
In practice, when a user clicks on a link, they are redirected to the advertiser's website. The http call to the advertiser page contains a field called http referer that indicates the origin of the click (the platform). The advertiser hence knows how the user landed on the the advertiser's site.
You can find out more here: https://en.wikipedia.org/wiki/HTTP_referer
The platform does not have any control on that.

---

> ### Author Rebuttal · Authors · 2025-07-31
>
> Thank you for your thoughtful review. We are grateful that you recognized our work as **"quite original"**, our PVM mechanism as **"simple and optimal"**, and that **its DSIC property addresses the cold-start problem**. We are especially grateful for your **crucial question regarding practical feasibility**, which has helped us realize the need to better clarify the modern industry context for this problem.  We also emphasize that our work is not just a theoretical issue. **As a leading company in the industry, we have direct insights from our proprietary data.**  Our responses to this are summarized in the table below, with detailed explanations to follow.
>
> ##  Summary of Responses and Planned Revisions
> |Question| Summarized Response|Revision|
> |-|-|-|
> | **Major Question**: The Practical Context and Platform Examples | Our work addresses a growing challenge faced by advertisers, such as app developers and e-commerce merchants, who use major platforms like Facebook, Google, and TikTok to direct users to **landing pages they do not control** (e.g., app stores, Amazon). For this group, the external-link verification you described is not feasible, and the traditional alternative—third-party redirects (MMPs)—is collapsing due to **new privacy policies** and **business concerns**. | We will update the **Introduction** to highlight the recent shift in tracking paradigms and the resulting challenges of platform misreporting. |
> | Real-Life Example | Two common examples of this scenario are **app downloads** and **e-commerce platforms**. | We will update the **Introduction** to better illustrate real-life scenarios. |
> | Ad Click Declaration and Verification Mechanisms | Advertisers match the **"Device Fingerprint"** collected after a user conversion (e.g., when a user first opens their app) with the click reports from the ad platform, which also include the fingerprint. **While this confirms that a user clicked, it does not allow the advertiser to independently verify when the click actually occurred, forcing them to rely on the platform's reporting time.** This information gap creates the opportunity for strategic manipulation, which our paper addresses. | We will update the **Introduction** to better illustrate the attribution procedure. |
> | Adapting the Mechanism for Multiple Clicks or Conversions per Platform | PVM can be applied to these scenarios **with minimal modification**. | - |
>
>
>
> ## Q1 Major Question: The Practical Context and Platform Examples
>
> In some scenarios—for instance, a brand like Nike directing ad traffic to its **own brand site**—verification via external links is straightforward.
>
> **Our work, however, focuses on the more complex and increasingly prevalent scenario where advertisers do not control the landing page**. A key example is a game developer advertising on TikTok whose landing page is the Apple App Store. In our model, the "platforms" are the major ad networks where these advertisers place their ads, primarily Facebook, Instagram, Google Ads, and TikTok Ads.
>
> The critical feature of this scenario is that the advertiser does not control the final landing page—key examples being app developers linking to the App Store or e-commerce merchants to a marketplace like Amazon. **This is precisely why the external-link verification method you described is not viable for them: they cannot deploy their own tracking pixels on these third-party platforms and thus have no independent way to verify the true click timestamp.**
>
> Historically, this verification gap for uncontrolled landing pages was filled by third-party **Mobile Measurement Partners (MMPs)**, such as AppsFlyer. These services track user interactions through **redirection**. When a user clicks on an ad, he/she are first directed through a MMP's URL, which logs the timestamp and metadata, before being redirected to the final landing page. This process allows the MMP to track the click and provide click data to the advertiser.
>
> However, recent changes in **platform policies** and evolving **advertiser priorities** have made redirect-based verification increasingly **difficult to implement**.
>
> **(1) Platform-Side Prohibition of Third-Party Redirection**
>
> Before 2020, MMPs like AppsFlyer and Adjust commonly used redirection to track ad clicks, capturing important data (e.g., the IDFA) before redirecting users to their landing pages. However, this practice has become increasingly restricted due to growing privacy concerns. **Regulations such as the General Data Protection Regulation (GDPR, 2018), China's Personal Information Protection Law (PIPL, 2021), and Apple's App Tracking Transparency (ATT, 2021) have significantly limited redirection paths.** These regulations primarily aim to protect user privacy and prevent third-party services from redirecting users to potentially unsafe websites. As a result, platforms like the Apple App Store, along with ad platforms such as Meta and Google Ads, now require users to be directed directly to the landing page, eliminating the ability to route user data through third-party servers. Consequently, advertisers can no longer rely on MMPs for click tracking.
>
> **(2) Advertiser-Side Constraints**
>
> Even in cases where redirects are not explicitly banned, advertisers are increasingly moving away from them due to several key factors:
>
> - **Protection of Proprietary Data**: Many advertisers are highly protective of their user data and reluctant to share it with external parties, even when such third-party solutions are not explicitly prohibited.
> - **Impact of Redirects on Conversion Rates**: Even when redirects are permitted, the growing prevalence of disruptive pop-up warnings has become a major concern. Many platforms, though not completely banning redirects, now display confirmation prompts (e.g., **"Are you sure you want to visit this link?"**) that create friction in the user experience, thereby discouraging users from completing their intended actions.
> - **High Costs of Redirect Solutions**: Whether using third-party MMPs or developing an in-house solution, implementing redirects incurs significant costs, making these solutions less attractive to advertisers.
>
> As a result of these developments, **the ability to use external links for  verification has been severely diminished**. Consequently, a large number of advertisers are left increasingly reliant on unverified, platform-reported data, making the strategic delay of timestamps a meaningful and urgent issue.  **Based on our unpublished research, this method of attribution through platform-reported click data is the dominant approach for app download ads, accounting for at least 70% of cases.**
>
> Furthermore, even if third-party verification were still possible, the traditional approach becomes a costly **"cat-and-mouse game"** between detection and fraud, as it does not address the platforms' inherent incentive to misreport data. **In contrast, PVM tackles the problem at its core by using mechanism design (DSIC), ensuring that honesty becomes the most profitable strategy for all involved parties.**
>
> ## Q2 Real-Life Examples
>
> Two common examples of this scenario are app downloads and e-commerce platforms:
>
> - **App Downloads**: When a user clicks on an ad for an app, he/she is sent directly to the App Store (e.g., Google Play or Apple’s App Store). Since platforms no longer allow redirects and the app store page is not controlled by the advertiser, the advertiser cannot use a MMP to track the exact timing of the click. This creates a situation where the advertiser cannot verify the attribution independently, and the platform can manipulate the timing of click reporting to affect attribution.
> - **E-commerce Platforms**: Similarly, when users click on ads for products on advertising platform (e.g., Facebook), they are sent directly to an e-commerce platform (e.g., Amazon). Just like in the app download scenario, since redirects are not allowed by Facebook or Amazon, MMPs cannot track clicks for advertiser.
>
> ## Q3 Ad Click Declaration and Verification Mechanisms
>
> When a user clicks an ad on a platform like Facebook or TikTok, the ad platform instantly captures a detailed **Device Fingerprint**. This includes key information such as the Device Advertising ID (e.g., IDFA or GAID, if permitted), IP address, and a User-Agent string. The platform then sends this data packet directly to the advertiser's server (maybe with some strategic delay).
>
> Later, if the user converts (e.g., by opening the newly downloaded app for the first time), the advertiser's SDK in the app collects the same type of Device Fingerprint from the device. By matching the fingerprint from this new conversion to the previously received click reports, the advertiser can then attribute the installation.
>
> **However, while this matching process can verify that a specific user clicked, it provides the advertiser with no way to independently verify when that click truly occurred.** This information asymmetry, and the resulting potential for strategic timestamp manipulation, is precisely the problem we address in our paper.
>
> ## Q4 Adapting the Mechanism for Multiple Clicks or Conversions per Platform
>
> PVM can be straightforwardly adapted to handle both scenarios. The key principle is that PVM operates on a per-conversion basis, considering the last click from each platform.
>
> - **Multiple Clicks**: When a single platform serves multiple ads to a user before one conversion, the standard last-click attribution principle is applied before PVM is invoked. Specifically, for each platform, only the timestamp of its final click prior to the conversion is considered a candidate for attribution. PVM then operates on this set of candidate last clicks (one from each participating platform).
> - **Multiple Conversions**: In scenarios where a platform contributes to multiple conversions, we apply PVM to each conversion independently. This ensures that each conversion event is attributed correctly.

---

> > ### Comment · Reviewer_Tb4U · 2025-08-04
> >
> > Hello, thank you for your answer.
> > I increased my score following your detailed answer, that clarified my misanderstanding.
> > I have one additional question: the paper suppose that each platform is characterised by a distribution.
> > But is this something that is observed in the data? Can you show a plot to back this claim?

---

> > > ### Author Response · Authors · 2025-08-04
> > >
> > > Thank you very much for increasing your score! We are glad our previous response **clarified the practical context** and are grateful for your further engagement with our work.
> > >
> > > Regarding your question about the empirical basis for assuming platform-specific distributions, **this assumption is well-supported by our data**.
> > >
> > > Actually, we have included a plot in the paper that illustrates this finding. Please see **Figure 2 in Appendix D**, which details how we get this distribution from real-world data and our numerical experiments.
> > >
> > > We’d be happy to continue the discussion if there are any further questions.

---

> > > > ### Comment · Reviewer_Tb4U · 2025-08-07
> > > >
> > > > Thank you for this clarification ! I have no further questions.

---

> > > > > ### Author Response · Authors · 2025-08-07
> > > > >
> > > > > Thank you again for your thoughtful engagement throughout the review process! We are pleased to **have addressed your concerns**. Your suggestion to **better illustrate the practical context of our work** is particularly valuable. We are confident that incorporating it into the final version will allow us to present our results more clearly and underscore that **designing incentive-compatible mechanisms for attribution** is an **emerging area** with significant **real-world need** and numerous **unexplored research avenues**!

---

### Official Review · Reviewer_nsc8 · 2025-07-02

**Clarity:** 2
**Significance:** 2
**Originality:** 3
**Rating:** 4
**Confidence:** 4

**Summary:**

This paper provides a new perspective on the advertising attribution problem by rethinking it through the lens of mechanism design, with a particular focus on the strategic aspects of reported click time. It reformulates the problem and theoretically demonstrates the limitations of the widely used Last-Click Mechanism (LCM). The authors propose a novel mechanism, the Peer-Validated Mechanism (PVM), which does not rely on the click time reported by the platform itself, and conduct a theoretical analysis of its properties. Additionally, the paper introduces a notion of fairness specific to this setting and provides an estimation for it.

**Questions:**

Please refer to "Weaknesses". Explanations for weaknesses 2, 3, and 4 are expected.

**Ethical Concerns:**

["NO or VERY MINOR ethics concerns only"]

**Final Justification:**

From the perspective of a one-shot game, the work is commendable. I give a borderline accept since the limitations cannot be ignored.

**Limitations:**

The elaboration of social impact is missing. The article designs a mechanism that might be applied in the real society. The ineffectiveness of self-reported information in increasing their own benefits for participants in this attribution mechanism may lead to irrational reporting, causing chaos.

**Paper Formatting Concerns:**

No formatting issues.

**Quality:**

3

**Strengths And Weaknesses:**

Strengths:
1. This perspective is novel — it reformulates the Ad attribution problem into a mechanism design problem from the angle of reported click time, under which the problem becomes clean.
2. Within the authors’ problem setting, the theoretical analysis of PVM is well-executed, including the elegant proof of the optimality of PVM for homogeneous platforms and a detailed derivation in the accuracy analysis.

Weaknesses:
1. This paper has some issues with its presentation.
- First, concerning citations: they should be placed appropriately where claims are made. For instance, citations supporting the mention of various methods in the second paragraph of the Introduction should be provided there, rather than all being clustered in the Related Work section.
- There is a discrepancy in Table 1: the reported accuracy for the LCM differs from the value given in Theorem 3.
- The wording around line 65 creates the impression that this is the first theoretical analysis of the problem. However, while you present a new modeling approach, theoretical analyses from other perspectives have been conducted previously.
- Regarding Figure 2.2, it seems rather pointless. If a figure is truly necessary, it would be better to include an introductory teaser figure at the beginning, explaining the basics of ad attribution and illustrating how the two mechanisms (LCM and PVM) work.
2. The problem formulation in this paper appears problematic.
- If advertisements use external links (as is now the mainstream practice), the ad provider can obtain the authentic timestamp at the moment of the click. Since this issue is easily resolvable, it raises doubts about the meaningfulness of this specific modeling assumption. The statement at line 33 ("a platform may delay the reporting...") likely requires a citation to substantiate the claim that platforms actually engage in this behavior.
- To my knowledge, the more common attacks in this context are click spamming and click injection. While click injection (which falls under the category of hijacking) is beyond the current scope of discussion, your model also fails to address the click spamming problem. For instance, a platform could report a click once when it occurs, and then report the same click again minutes later. This behavior wouldn't reduce its own revenue but would unfairly diminish the attributed revenue of other platforms (this also relates to normalization issues, which will be discussed later).
3. Problem Formulation: Normalization Issue.
- In Section 2.3, the constraint in Equation (3) is somewhat unsatisfying. This expectation-based constraint is arguably too weak. Crucially, unlike the LCM, it does not guarantee that the total credit sums to exactly 1 for each individual conversion. Under this condition, if the advertiser’s total budget is fixed, then any action by a platform that reduces the attributed revenue of others would inherently increase its own revenue. Consequently, the Peer-Validated Mechanism appears not necessarily DSIC in this specific sense.
4. The PVM exhibits excessive reliance on the conversion time distribution function $F_i(t)$.
- In the PVM design, $\alpha$ and $\beta$ are critical parameters whose values entirely depend on accurate knowledge of $F_i(t)$. This requirement for $F_i$ constitutes an overly strong informational assumption. Although the authors claim it is a "commonly known distribution" at line 116, this distribution is fundamentally unknowable under the paper’s core premise that "advertisers cannot obtain authentic click times".
- The computation of $\alpha$ and $\beta$ is highly sensitive to the precision of $F_i(t)$ due to its multiplicative nature. Coarse estimates of $F_i(t)$ may be insufficient for reliable implementation.
- Moreover, the $F_i$ across platforms are likely correlated, yet the paper models each platform’s click time using a univariate distribution over time without justification. This interdependence remains unaddressed.
- Conversely, if one does possess the prior distributions $F_i$ and knowledge of which platforms participated in a conversion, could an allocation rule be designed that directly assigns credit (disregarding reported timestamps entirely) while preserving desirable incentive properties?

---

> ### Author Rebuttal · Authors · 2025-07-30
>
> Thank you for your detailed review. We sincerely appreciate your recognition of our **novel perspective**, **well-executed theoretical analysis**, and the **elegant proof**. We will address the typos you mentioned in Weakness 1 and have summarized our overall responses to the other concerns below, with detailed replies provided further on. **Apologies for the brevity, but due to space limitations, a more detailed rebuttal on strategic feasibility can be found in our responses to Reviewer Fr9B and Tb4U, while the discussion on the independence of $F_i$ is addressed in our response to Reviewer nsc8.**
> ## Summary of Responses and Planned Revisions
> | Question | Sub-question| Summarized Response | Planned Revisions |
> |-|-|-|-|
> | W2 Problem Formulation | Strategic Delayed Feasibility | We focus on the scenario where the landing page is out of the advertiser's control (e.g., app stores). The strategic feasibility for platforms to delay click reports arises from **the infeasibility of third-party verification via MMP redirection**, due to **a shift in tracking practices since 2020**, driven by **new privacy regulations** (e.g., GDPR, ATT) and **advertiser concerns** (e.g., protection of proprietary user data). | Revise **Introduction** to clarify problem context and add industry citations.  |
> |  | Other Click Attacks  | PVM removes the incentive for a rational agent to "spam", as this action offers no personal gain under our utility model.|-|
> | W3 Normalization Issue | Expectation-based Constraint| This is a deliberate choice reflecting **the practical goal of long-term budget control** while allowing for mechanism design flexibility. |-|
> | | DSIC Claim| The DSIC holds under the standard self-interested agent model $(u_i=x_i)$. The insightful point about a fixed budget is **a compelling future research direction exploring **repeated games with externalities**.                                                             | We will add a discussion to **the Future Work part in Conclusion** on modeling the problem as a repeated game with externalities. |
> | W4 Prior Issue | Knowledge of Prior  |PVM's DSIC property addresses the "cold-start” problem of unknown or inaccurate priors by allowing platforms to begin with arbitrary priors, collect honest data (from platforms) over time, and gradually learn accurate priors.| We will revise the **Model and Preliminaries** section to better explain the benefits of DSIC, ensuring it is clearer to a broader audience.|
> |   | Independence Assumption| **Our PVM framework is extensible to the general correlated case, preserving its core desirable properties.** We focused on the independent case because it is **sufficient** to present our foundational results, while also offering maximum **clarity** and the strongest analytical bounds.     | We will add a discussion to the **Conclusion** to clarify our independence assumption. |
> |    | Possible Prior-Only Mechanism | While the prior-only mechanism you suggest is incentive-compatible, it is **suboptimal** in accuracy.  | -   |    |
>
> ## W2:  Problem Formulation
> This is not just a theoretical issue. **As a leading company in the industry, we have direct insights from our proprietary data.**
> ### (W2.1) Strategic Delayed Feasibility
>
> The user conversion process could be classified into 2 scenarios based on the landing page:
> - **Advertiser-controlled landing pages**: For example, when a company runs ads on Facebook and directs users to its own website. In this case, the advertiser can easily record the true click timestamp. **However, this is not the focus of our paper**.
> - **Advertiser-uncontrolled landing pages**: This includes cases like app developers advertising their apps (with the landing page being an app store) or merchants promoting products on third-party platforms (e.g., Amazon). Here, advertisers cannot record click timestamps by themselves because they don’t control the landing page.
>
> In cases with advertiser-uncontrolled landing pages, advertisers often use **Mobile Measurement Partners (MMPs)**, like AppsFlyer, to track user interactions. The process involves **redirecting** users through a MMP's URL before reaching the landing page. When a user clicks on an ad, he/she first goes to the MMP's URL, which logs the timestamp and metadata, then is redirected to the final landing page. This allows the MMP to independently track the click and provide the data to the advertiser.
>
> **However, recent changes in platform policies and advertiser priorities have made redirect-based verification harder to implement**:
>
> - **Platform-Side Prohibition on Third-Party Redirection**: Regulations like GDPR (2018), China’s PIPL (2021), and Apple’s ATT (2021) limit redirection paths to protect user privacy and prevent unsafe redirects. As a result, many app stores and ad platforms now require users to go directly to landing pages, cutting out third-party servers.
> - **Advertiser-Side Constraints**: (**Data Privacy**) Many advertisers are increasingly protective of their proprietary user data and are unwilling to share it with external parties like MMPs. (**Redirect Impact on UX**) Even when redirects are allowed and data privacy is not a concern, many platforms display warning messages—like **“Are you sure you want to visit this link?”**—that disrupt the user experience and hurt conversion. (**Cost of Redirect Solutions**) Implementing redirects, whether through MMPs or in-house systems, is costly, making it less attractive.
>
> Due to these changes, external-link-based verification is no longer viable for many advertisers, leaving them dependent on unverified platform-reported data.   **Based on our unpublished research, this method of attribution through platform-reported click data is the dominant approach for app download ads, accounting for at least 70% of cases.**
>
> Furthermore, using third-party verification becomes a costly **"cat-and-mouse game"** of detection versus fraud, as it doesn’t address the underlying platform incentive to misreport. **In contrast, PVM addresses the root cause by using mechanism design (DSIC) to make honesty the most profitable strategy.**
>
> ### (W2.2) Other Click Attacks
> PVM resists click spamming for the same reason as the normalization issue. In LCM, platforms are incentivized to spam by being rewarded for the last reported click. However, under PVM, platforms have no such incentive, as their reward depends on peer reports, not their own. Therefore, spamming has no impact on their payoff, regardless of how much they spam.
>
> ## W3: Normalization Issue
>
> ### (W3.1) Expectation Constraint
> The expectation-based normalization ($E[\sum x_i]\leq 1$) was a deliberate one, motivated by **the practical objective of ensuring the advertiser's budget is respected on average over many conversions.** This approach provides the necessary **long-term budget control that advertisers require**, while allowing for the flexibility needed to design a powerful incentive-compatible mechanism.
> ### (W3.2) DSIC Claim
> PVM is DSIC under our standard utility model ($u_i = x_i$), where agents are rational and self-interested, meaning their utility depends only on their own payoff. In this model, platforms are indifferent to the payoffs of their peers.
>
> The idea of a fixed advertiser budget is insightful, as it leads to a new research direction: **a repeated game with externalities**. In this setting, platforms would be incentivized to hurt their peers’ performance to capture a larger share of the fixed budget. **This is a sophisticated challenge, as such externalities can compromise the incentive properties of even canonical truthful mechanisms like VCG.** However, we believe **it is a compelling direction for future research, and it further convinces us that strategic behavior in ad attribution is a rich field with significant value**.
>
> ## W4: Click Time Distribution Issues
>
> ### (W4.1) Knowledge of Prior
> **The "unknowable prior" is exactly why DSIC is important.**  PVM's DSIC property solves the 'cold-start' problem by allowing platforms to start with arbitrary priors. Since DSIC ensures truthful reporting, the data collected (from platforms) over time is honest, allowing advertisers to learn the correct distribution and improve their priors.
>
> ### (W4.2) Independence Assumption
> Our PVM framework **can be extended to settings with correlated click-time distributions**:
> - **Mechanism Modification**: Instead of a single threshold ($\alpha_i$), the mechanism defines a multi-dimensional "acceptance region" $D_i$ based on peer reports $t_{-i}$. This region is constructed by a greedy algorithm that adds the peer-report outcomes with the highest probability that platform $i$ was the true last click. A platform receives credit if its peers' reports fall within this region.
> - **Impact on Results**: Our conclusions in the homogeneous setting remain unchanged, as the proof is based on the symmetric structure. In the heterogeneous setting, LCM's performance remains unchanged, as its accuracy is already near zero. The main change is PVM's accuracy guarantee, which becomes a weaker $1/n$ lower bound.
>
> Although we can extend PVM to correlated settings, the focus of this paper is on introducing the peer-validation framework and presenting two key contributions: **the first theoretical model for strategic misreporting** in the "no-redirect" paradigm, and **a peer-validated mechanism** as a robust solution. **The independence assumption is sufficiently to fully characterize the problem, while also making PVM’s superiority clearer through a simpler mechanism and stronger analytical bounds.** We will discuss this choice in the Conclusion to strengthen the paper.
>
> ### (W4.3) Possible Prior-Only Mechanism
> A prior-only mechanism is DSIC but suboptimal for our objective (Equation 4). In contrast, **PVM is the optimal DSIC mechanism in the homogeneous setting and superior to LCM**, as shown both theoretically and experimentally.

---

> > ### Comment · Reviewer_nsc8 · 2025-08-04
> >
> > Thank you for your clarification. The concern regarding problem formulation has been addressed.
> >
> > However, the 'cold-start' justification seems problematic. Even if the problem satisfies DSIC under the specified constraints, it must be noted that this setting only eliminates agents' incentive to report incorrect times. It does not guarantee they will report the correct time. Agents who understand the mechanism design may deliberately adhere to a distribution when reporting time (though such reporting may depend on external information, which could be obtainable). This allows them to handcraft their reported distribution to increase utility. In the 'cold-start' setting, distributions are not a priori available; they are incrementally learned through repeated reporting. Crucially, since the evolving distribution influences utility, the problem transforms into a distinct (and substantially more complex!) scenario where the distribution itself becomes a variable gradually estimated from each reporting event. Please confirm whether my understanding above is correct.
> >
> > Two very minor additional questions, answer as you see fit: Your original submission contained no phrasing close to 'cold-start' – is this an interpretation you conceived for the rebuttal? And, as a leading company in the industry, is the mechanism disclosed to all parties in actual implementation?

---

> > > ### Author Response · Authors · 2025-08-05
> > >
> > > Thank you for confirming that **your concern regarding the problem formulation has been addressed**. We appreciate the opportunity to use your follow-up question to further clarify **our contribution** and its relationship to **the broader landscape of mechanism design**.
> > > ## 1. The Strategic Distribution Manipulation Problem You Mentioned in Paragraph 2
> > > We understand your concern regarding potential distribution manipulation.  To address this, we will first clarify the precise scope of our problem setting—**the one-shot attribution game**—and contrast it with the different strategic considerations of a **repeated game**. Then, we will discuss the practical barriers that inhibit a platform's ability to manipulate its distribution.
> > >
> > > **Our analysis is grounded in the one-shot game model.** The utility function defined in our paper $U_i(\tau_i, \tau_{-i}) = \mathbb{E}_t[x_i(t + \tau)]$ ,  formally models a platform's objective as maximizing its expected credit from a single, immediate conversion event. This framework, therefore, analyzes the behavior of rational agents focused on a per-instance payoff, rather than on long-term strategic considerations such as influencing the advertiser's future priors. Therefore, a more accurate statement of sentence 2&3 in your second paragraph is, "The mechanism guarantees that the platform, under the specific objective  $U_i$, will report the correct time." This is a crucial guarantee that the industry-standard LCM fails to provide, leading to its poor performance in accuracy and fairness.
> > >
> > > **Indeed, the strategic manipulation of priors you describe correctly identifies the dynamics of a repeated game.** This challenging and under-exploring frontier represents a promising area for future research, not only in ad attribution but across many areas of mechanism design, including auctions and resource allocation. **Even for canonical DSIC mechanisms like Myerson mechanism, ensuring incentive compatibility in dynamic settings is a challenge whose solutions remain a largely underexplored research frontier.** Additional insights on this topic can be found in:
> > > -  A Game-Theoretic Analysis of the Empirical Revenue Maximization Algorithm with Endogenous Sampling. (Deng et al.  NeurIPS 2020) ,
> > > - The Price of Prior Dependence in Auctions. (Tang et al. EC18)
> > > - Budget-Constrained Auctions with Unassured Priors: Strategic Equivalence and Structural Properties. (Chen et al. WWW 2024)
> > >
> > > Furthermore, from a practical standpoint, systematically manipulating the prior distribution presents significant challenges.  **The formal definition of the PVM framework specifies the attribution rule but not the methodology for estimating priors; this is an implementation detail external to the mechanism's public commitment.** Consequently, rather than naively fitting only the reported data, a sophisticated advertiser can build a more robust estimation by incorporating a wealth of exogenous signals. These can include a platform's ad volume, the performance of its peers, and ad creative quality—factors that are largely beyond any single platform's strategic control.
> > >
> > > ## 2. Two Minor Questions
> > > - **cold-start interpretation** : We want to state clearly that this 'cold-start' interpretation was not conceived for the rebuttal. In mechanism design, employing a DSIC mechanism is the classical approach to ensure practical implement ability, precisely because of its foundational ability to elicit truthful data. In keeping with the conventions of theoretical mechanism design, our original submission did not elaborate on this practical motivation in order to maintain a strict focus on the core contribution--- the formal model of strategic behavior and the rigorous analysis of our proposed solution. We initially classified the 'cold-start' problem as an implementation consideration rather than a part of the core theory. Nevertheless, we agree that this context is valuable to the reader and will incorporate this discussion into our revision. We appreciate the suggestion.
> > > - **mechaniam disclosed**: Regarding mechanism disclosure, **it is the advertiser who chooses and commits to the attribution mechanism for all participating platforms.** When an advertiser adopts a DSIC mechanism like PVM, rational platforms are incentivized to report truthfully. From the perspective from a leading platform, we advocate for the adoption of PVM precisely for this reason. The alternative, LCM, creates a destructive equilibrium: even a platform that prefers to report honestly may be forced into defensive misreporting to prevent severe losses if its competitors are strategic. PVM resolves this dilemma, fostering a more stable and trustworthy ecosystem.
> > >
> > > We apologize for the incorrect rendering of the mathematical formula. Despite multiple attempts, we were unable to compile it correctly within the OpenReview system. We’d be happy to continue the discussion if there are any further questions.

---

> > > > ### Comment · Reviewer_nsc8 · 2025-08-07
> > > >
> > > > Thank you for your detailed reply. From the perspective of a one-shot game, your work is commendable, and I am willing to raise the rating. Please delineate the boundaries of this work to avoid misunderstandings in the final version.

---

> > > > > ### Author Response · Authors · 2025-08-07
> > > > >
> > > > > Thank you for raising your score and for your active engagement throughout this discussion! We sincerely appreciate it.
> > > > >
> > > > > Your detailed and thoughtful review has provided a clear roadmap for refining the final version of our paper — especially regarding its **practical context**, **limitations**, and **future directions**. In addition, your comments have already inspired our thinking for a follow-up project. We will carefully revise the final version with these insights in mind.
> > > > >
> > > > > We believe that mechanism design for attribution is a field with significant practical value and numerous unexplored research avenues. **It offers a complementary approach to the sophisticated attribution philosophies and the highly costly detection or supervised techniques currently prevalent in the field, and we hope our paper will encourage further exploration in this area!**

---

> > > ### Author Response · Authors · 2025-08-07
> > >
> > > Dear Reviewer nsc8,
> > >
> > > We hope this message finds you well. As the rebuttal phase is nearing its end (with less than two days remaining), we would like to kindly follow up regarding our response to your comments. We would greatly appreciate it if you could let us know whether our reply has addressed your concerns adequately, or if there are any remaining questions or clarifications you'd like us to provide.
> > >
> > > Thank you again for your time and effort in reviewing our work.
> > >
> > > Best regards,
> > >
> > > Authors

---

### Official Review · Reviewer_jUsn · 2025-07-03

**Clarity:** 3
**Significance:** 3
**Originality:** 3
**Rating:** 5
**Confidence:** 4

**Summary:**

The paper studies a theoretical model for ad attribution. The goal if to attribute a conversion to the last click before this conversion, but to do this, we need to rely on the reports of click times from a number of self-interested platforms. The natural and commonly-used LCM mechanism that simply attributes the contribution to the last reported click is not incentive compatible: platforms might have an incentive to delay their report. The paper proposes a dominant-strategy incentive-compatible mechanism called PVM for this problem, assuming known priors on the time between clicks on each platform and the conversion (and if I understand correctly, assuming that these time delays are independent). The mechanism credits a platform if the other platforms report times that are earlier than some thresholds, ignoring the platform's own report (beyond checking that the click is reported before the conversion). Obviously if platforms report their times truthfully, the LCM is perfectly accurate in picking the last click and PVM is sometimes inaccurate. But interestingly, the paper shows that if platforms act strategically, then in an equilibrium, LCM can be much less accurate in picking the last click than PVM. In fact, they show that PVM is the most accurate incentive compatible mechanism. The paper also shows a fairness property of PVM and evaluates the accuracy and fairness of PVM compared to LCM in numerical experiments for small sizes.

**Questions:**

* Do you need the assumption that F_i's are independent?
* Why do thresholds always exist? Are they unique? What happens if they don't exist or are not unique?
* Can anything be said in a setting without priors or with inaccurate priors?

**Ethical Concerns:**

["NO or VERY MINOR ethics concerns only"]

**Limitations:**

yes

**Quality:**

3

**Strengths And Weaknesses:**

Strengths:
+ Interesting and novel theoretical solution for a non-standard mechanism design problem.
+ It is quite interesting (and perhaps surprising) that the authors can show the optimality of PVM wrt accuracy.
+ From a technical point of view, the paper is significantly novel and is not a simple application of known methods.

Weaknesses:
- Assumptions of the model are not properly discussed. Mainly:
  - It is assumed that there are known priors commonly known by everyone. What if that is not the case?
  - It seems that the paper assumes that the F_i distributions are independent, but this is not clearly stated. Also, I don't think this is a reasonable assumption, since the conversion delay times on different platforms are clearly correlated with the conversion time and therefore are intercorrelated. I'd understand if an unrealistic assumption like this might be necessary to obtain theoretical results, but this needs to be properly discussed.
  -  Do thresholds alpha_S^i always exist? Are they unique?  I guess some form of regularity might be necessary for the existence and uniqueness of these thresholds.
- While the results are interesting from a theoretical point of view, I don't expect any practical application (mainly because of the unrealistic assumptions).
- Minor: the title of the paper is a bit misleading. The paper does not go beyond the last-click attribution. It proposes a better mechanism for last-click attribution in a strategic environment.

---

> ### Author Rebuttal · Authors · 2025-07-30
>
> We sincerely thank you for your constructive feedback! We are grateful that you recognized our work's **significant technical novelty** and the **"surprising" accuracy optimality** of our PVM mechanism. For the concerns you raised about our model's assumptions, our responses are summarized below, with detailed replies provided further on.
>
> ## Summary of Responses and Planned Revisions
>
> | Question | Summarized Response | Revision |
> |-|-|-|
> | Need for Independent Assumption | **Our PVM framework is extensible to the general correlated case, preserving its core desirable properties.** We focused on the independent case because it is **sufficient** to present our foundational results, while also offering maximum **clarity** and the strongest analytical bounds. | We will add a discussion to the **Conclusion** to clarify our independence assumption. |
> | Threshold Existence and Uniqueness | The thresholds' existence and uniqueness are guaranteed under standard regularity conditions on the priors, and the mechanism is easily adapted for any theoretical edge cases. | We will clarify in a new **footnote** along with the handling of theoretical edge cases |
> | Settings Without Priors or with Inaccurate Priors | PVM's DSIC property addresses the **"cold-start”** problem of unknown or inaccurate priors by allowing platforms to begin with arbitrary priors, collect honest data (from platforms) over time, and gradually learn accurate priors. | We will revise the **Model and Preliminaries** section to better explain the motivation for choosing a DSIC mechanism, ensuring its benefits are clear to a broader audience. |
> | Practical Value Concern | The practical value lies in a twofold contribution: (1) providing a **theoretical diagnosis** of the root cause of strategic reporting delays, and (2) offering **a concrete design principle for future attribution mechanisms.** | We will expand the **Future Work part in Conclusion** section  to discuss the practical implications of our work and suggest paths toward real-world implementation. |
>
> ## Q1: The Need for Independence Assumption
>
> We apologize if the statement of our independence assumption was not sufficiently prominent. We do state it on line 117, but we will ensure it is highlighted more explicitly.
>
> **Our PVM framework can be extended to settings with correlated click-time distributions.** The core peer-validation principle is preserved, ensuring the mechanism remains DSIC, but the validation rule is generalized:
>
> - **Mechanism Modification**: Instead of using a single threshold ($\alpha_i$), the mechanism would define a multi-dimensional "acceptance region" $D_i$ for the vector of peer reports $t_{-i}$. To maximize accuracy, this region $D_i$ is constructed by a greedy algorithm: we iteratively add the peer-report outcomes $t_{-i}$ that yield the highest posterior probability that platform i was the true last click (i.e., $P(t_i > \max\{t_{-i}\} | t_{-i})$ is highest). This process continues until the total probability of $t_{-i}$ falling within this region equals the prior probability that i is the last click ($\beta_i$). A platform $i$ receives credit if and only if its peers' reports $r_{-i}$ fall within this precomputed region $D_i$.
>
> - **Impact on Results**: Under this modification, our conclusions in the homogeneous setting remain unchanged, as the proof is based on the symmetric structure. In the heterogeneous setting, the conclusions for LCM also remain unchanged, as its accuracy and fairness have already approached zero and thus cannot degrade further. The primary change is PVM's performance in the heterogeneous setting, where the accuracy guarantee becomes a weaker (but still valuable) $1/n$ lower bound. While this is still strictly superior to LCM's worst-case accuracy of 0, it lacks the tighter bound derived under the independence assumption.
>
> While our PVM framework can be extended to the correlated case, our primary goal with this paper was to introduce our peer-validation framework in the clearest possible setting. We believe our paper makes a twofold contribution: providing the first theoretical model for strategic misreporting under the new "no-redirect" paradigm, and introducing our novel peer-validated mechanism as a robust solution. **The independence assumption provides an ideal setting to demonstrate these contributions, as it is sufficient to fully characterize the strategic problem while showcasing PVM's superiority with a simpler mechanism, more elegant proofs, and stronger analytical bounds.** Intuitively, this can be viewed from a backward-looking perspective: given a conversion, the advertiser forms an estimate for each platform's time lag, and the uncertainties in these separate estimates are modeled as independent. We sincerely thank the reviewer for raising this point, as it has helped us realize the importance of making this reasoning explicit. We will add a discussion on this methodological choice to the Conclusion to make the paper stronger.
>
> ## Q2: The Existence and Uniqueness of PVM Thresholds
>
> Thank you for this excellent question regarding the technical conditions for our thresholds. The thresholds $\alpha_i$ exist and are unique under standard regularity conditions, and the mechanism can be straightforwardly adapted for edge cases.
>
> As defined in our paper, the threshold $\alpha_i$ is the solution to $G_i(\alpha_i) = \beta_i$, where $G_i(t) = \Pi _{j\neq i} F_j(t)$ is the cumulative distribution function (CDF) of the maximum click time among platform $i$'s peers, and $\beta_i$ is the prior probability that $i$ is the true last click.
>
> - **Standard Case (Existence and Uniqueness)**: If $G_i(t)$ is continuous and strictly increasing. In this standard case, for any $\beta_i$ in $(0,1)$, a unique solution $\alpha_i$ is guaranteed to exist by the Intermediate Value Theorem.
>
> - **Special Case 1 (Flat CDF)**: If $G_i(t)$ has a flat region, and $\beta_i$ falls within this flat range, the threshold $\alpha_i$ would not be unique. However, a flat region in $G_i(t)$ implies that the probability of the maximum peer click time occurring within that specific interval is zero ($P(\max\{t_{-i\}} \in [a,b]) = 0$). Therefore, any threshold $\alpha_i$ chosen within this flat interval will yield the exact same attribution outcome. The choice is arbitrary and has no impact on the mechanism's performance. We acknowledge this possibility in our proof sketch for Lemma 1.
>
> - **Special Case 2 (Discontinuous CDF)**: In the rare event that $G_i(t)$ could have a jump discontinuity at a point $\theta$ such that $G_i(\theta-) < \beta_i < G_i(\theta)$, a single threshold $\alpha_i$ would not exist. This scenario implies a non-zero probability that $\max\{t_{-i}\} = \theta$. The mechanism can be modified to handle this by using a probabilistic assignment: if $\max\{\boldsymbol{t}_{-i}\} = \theta$, we assign credit to platform $i$ with a specific probability $p$ such that the expected attribution remains $\beta_i$. However, we consider this case to be of limited practical relevance, as click times are typically modeled as continuous random variables.
>
> ## Q3: Settings Without Priors or with Inaccurate Priors
>
> We see the settings without priors or with inaccurate priors as a **cold-start problem**. This is precisely why we pursued a DSIC mechanism--- **Learning Priors via a Virtuous Cycle**.
>
> Even if an advertiser starts with inaccurate priors (or no priors at all), PVM's DSIC property solves the ''cold-start'' problem by allowing platforms to start with arbitrary priors. Since DSIC ensures truthful reporting, the data collected (from platforms) over time is honest, allowing advertisers to learn the correct distribution and improve their priors. This allows them to iteratively learn and refine their estimates of the true click-time distributions over time.
>
> **This creates a powerful virtuous cycle:** truthful data leads to more accurate priors, which in turn makes the PVM mechanism more precise and effective. **In contrast, a non-DSIC mechanism like LCM offers no such path**; it consistently provides biased data, making it impossible for the advertiser to ever learn the true distributions. Under LCM, platforms are instead incentivized to continuously develop new ways to game the system, leading to a negative spiral.
>
> ## Concern: Practical Value of Our Theoretical Work
>
> In response to the question of practical application, we believe our theoretical work offers value on two distinct levels: **(1) providing a foundational diagnosis of the root cause of strategic misreporting in the new "no-redirect" conversion paradigm, and (2) offering a concrete design principle ("peer-validation") for future systems.**
>
> First, our work frames the issue of misreporting. We show it is not a simple monitoring problem to be solved with better detection, but rather a fundamental incentive flaw in the widely-used Last-Click Mechanism. This shifts the focus of any practical solution away from a costly **"cat-and-mouse game"** of enforcement and towards the core need for incentive-compatible design.
>
> This diagnosis leads to our second insight: a concrete and transferable design principle of "peer-validation." This principle is highly practical. **For instance, in a modern system using a neural network for attribution, our key engineering guideline to ensure truthfulness (DSIC) is that a platform's credit must be calculated without using its own reported timestamp as an input feature.** This is a clear, actionable takeaway for practitioners developing the next generation of robust, ML-based models.
>
> We believe this work serves as a valuable starting point for designing trustworthy attribution mechanisms in the new "no-redirect" paradigm, and we hope it will inspire more research in this critical and evolving area.

---

> ### Author Response · Authors · 2025-08-07
>
> Dear Reviewer jUsn,
>
> Thank you again for your effort reviewing our paper, providing positive assessment and valuable feedback. Since the discussion period is ending soon, we would like to follow up to see if our rebuttal has addressed your concerns and if there are any other questions. We would be happy to provide any further clarification. Thank you very much.
>
> Sincerely,
>
> Authors

---

### Official Review · Reviewer_Fr9B · 2025-07-03

**Clarity:** 3
**Significance:** 3
**Originality:** 3
**Rating:** 4
**Confidence:** 2

**Summary:**

This paper studies attribution in multi-platform advertising where platforms may strategically delay impression timestamps to game credit assignment. It introduces the Peer-Validated Mechanism (PVM), which is dominant-strategy incentive compatible (DSIC), provably fair, and optimal among threshold-based mechanisms. Theoretical results are thorough and correct, and extensive experiments using real-world click data demonstrate PVM’s superiority in accuracy and fairness over Last-Click attribution.

**Questions:**

1. Strategic Feasibility in Practice: How common is timestamp manipulation in practice? Can you clarify when platforms are able (and willing) to delay impression timestamps, especially given common audit mechanisms or advertiser-side control?
2. Learning Dynamics: Would LCM perform better if platforms adaptively learned their best delay strategies over time (e.g., via regret minimization)? Could PVM be robust in a repeated-game setting?
3. PVM Threshold Implementation: How complex is threshold computation in PVM for large n? Can you comment on runtime or whether thresholds can be precomputed in practice?
4. Advertiser Utility: Would incorporating advertiser-side objectives (e.g., conversion accuracy or ROI) affect the relative performance of LCM and PVM? Have you considered joint optimization over platform and advertiser utilities?
5. Noise Robustness: Your simulations assume precise timestamps. How would PVM’s performance change under real-world noise: jittered impressions, delayed reports, or soft conversions?

If you address (1), (3), and (5), it would strengthen my score — especially if you can show that PVM remains practical and truthful even under noisy, partial, or delayed observations.

**Ethical Concerns:**

["NO or VERY MINOR ethics concerns only"]

**Limitations:**

* Assumption Realism: The core setting assumes platforms can delay timestamp reports (but not report earlier). This is valid for theoretical modeling, but in practice, such manipulation is often restricted. The paper should explicitly discuss whether such strategic timing is realistic under ad industry audit standards, MMPs, or advertiser-enforced attribution windows.
* Limited Economic Impact Discussion: The paper proves fairness in attribution, but doesn’t discuss whether that leads to better advertiser outcomes (e.g., campaign optimization) or negative impacts (e.g., over-crediting fast-reporting platforms).

**Quality:**

3

**Strengths And Weaknesses:**

### Strengths:
* (S1) The paper presents a rigorous and well-structured treatment of attribution in multi-platform ad settings, modeling strategic timestamp misreporting and proposing the Peer-Validated Mechanism (PVM) as a dominant-strategy alternative to the commonly used Last-Click Mechanism (LCM).
* (S2) The experimental section (Sections 5 and Appendix D) is a major strength. It uses real-world click-timing data from four anonymized platforms, and validates PVM’s superiority in both accuracy and fairness across homogeneous and heterogeneous settings,

### Weaknesses:
* (W1) Timestamp Manipulability:  Platforms don’t usually control the raw timestamp of an impression — those are logged by an ad server or measurement pixel. Misreporting impression times could violate platform policies or audit trails. Thus, the assumption that platforms can delay reporting of impression times to strategically influence attribution (without audit consequences) is theoretically coherent but may be limited in practice. I recommend clarifying the conditions under which such misreporting is feasible, and discussing how this maps to real-world ad attribution ecosystems.
* (W2) Unsupported claims: Readers unfamiliar with the domain lack guidance on where to look for background, as introduction does not contain any reference (apart from the related work section) and has a number of unsupported claims.  For example, “Advertisers interact with a small number of platforms”, “platforms may strategically misreport to gain greater attribution”, etc.

---

> ### Author Rebuttal · Authors · 2025-07-29
>
> Thank you for the positive feedback and constructive suggestions. We appreciate that you found the **rigor** and **well-structured theoretical results** valuable, as well as the **experiments using real-world data** that demonstrated PVM's superiority in accuracy and fairness. For the concerns you raised, we have summarized our overall responses below, with detailed replies provided further on.
>
> ## Summary of Responses and Planned Revisions
>
> | **Question**| **Summarized Response**| **Revision**|
> |-|-|-|
> | Strategic Feasibility in Practice | Strategic feasibility arises because advertisers are often blind in funnels they don't control (e.g., direct-to-app-store installs), forcing a near-total reliance on platform-reported data as privacy regulations (like ATT) have dismantled traditional audits. | We will update the **Introduction** to highlight the recent attribution paradigm shift and the resulting platform misreporting challenges. |
> | Learning Dynamics| LCM's performance does not improve with adaptive learning. PVM is robust in repeated games due to its DSIC property. |-|
> | PVM Implementation| PVM's thresholds can be **precomputed offline**, and an online runtime complexity of $\mathcal{O}(n^2)$ per conversion. But ad attribution is typically an offline batch process without stringent latency requirement| We will update the **PVM section** to include an analysis of its computational complexity.|
> | Advertiser Utility| A full, joint-optimization game that models both advertiser and platform utilities while integrating the attribution and budget allocation processes represents **a compelling direction for future research**. This convinces us that the field of ad attribution is rich with open questions, and we hope our foundational work can serve as a catalyst for such studies. | We will update the **Conclusion** to point towards joint optimization as a compelling direction for future work.|
> | Noise Robustness| **PVM is robust to real-world noise.** The DSIC property eliminates strategic manipulation, and operational noise (e.g., jitter) can be managed with standard data-cleaning techniques. | We will expand the **Conclusion** to include a discussion on PVM's robustness to real-world noise. |
>
> ## Q1: Strategic Feasibility in Practice
> This issue is not only feasible but increasingly prevalent, particularly in common advertising funnels where advertisers, lacking  third-party audit capabilities, must rely on the self-reported data from the platforms. **As a leading platform in the industry, we have direct insights into this challenge.**
>
> ### A Prime Scenario: Direct-to-App-Store Advertising
> A prime and highly common example is the direct-to-app-store advertising funnel. In this scenario:
> - A user clicks an ad on a major platform (e.g., Meta, Google, TikTok).
> - The user is taken directly to the Apple App Store or Google Play Store, with no intermediate redirection.
> - The user installs the app. The advertiser (e.g., a game developer) receives a notification of the install from Apple/Google, but has no independent way to know which specific ad click led to it.
>
> **According to unpublished research data from our company, this method of attribution through platform-reported click data is the dominant approach, accounting for at least 70% of app download ads.**
>
> In these cases, the platform becomes the sole source of truth for the click timestamp. This dependency creates a clear and widespread opportunity for the strategic manipulation our paper addresses.
>
> ### Why Traditional Audits Are Becoming Infeasible?
>
> The prevalence of these "blind-reliance" scenarios has been exacerbated by the very industry shifts, which have dismantled traditional verification methods:
>
>  **(1) The Decline of Third-Party Redirection**: Prior to 2020, Mobile Measurement Partners (MMPs) could use redirects for verification. However, this is now heavily restricted. New privacy regulations (e.g., China's Personal Information Protection Law (PIPL) and Apple's App Tracking Transparency (ATT) )  prohibit or penalize such redirects to protect user data and experience. Major app stores and ad platforms now require a direct path to landing page, eliminating the crucial touchpoint for independent, real-time auditing.
>
>  **(2) Advertiser-Side Constraints**
> Even when technically possible, advertisers themselves are moving away from complex tracking for key business reasons:
> - **Protection of Proprietary Data:** Advertisers are increasingly reluctant to share their user data with external parties.
> - **Degraded User Experience:** Redirects often introduce disruptive pop-up warnings (e.g., "Are you sure you want to visit this link?"), increasing user drop-off.
> - **High Cost:** Implementing and maintaining robust redirect solutions is expensive.
>
> These changes have created a widespread environment where platforms have the unilateral ability to report timestamps without facing easy, real-time detection from advertisers. The willingness to exploit this reporting ability is driven by the powerful financial incentives of the LCM.
>
> Therefore, while some forms of historical or sample-based auditing might exist, they represent a costly and imperfect **"cat-and-mouse game"**. Our work proposes a more fundamental solution. By introducing the PVM, we leverage mechanism design to create an incentive-compatible system where truthful reporting becomes the dominant, most profitable strategy. **This obviates the need for external audits by fixing the problem at its source.**
>
> ## Q2: Learning Dynamics
>
> **For LCM, adaptive learning dynamics would not improve its performance.** While learning processes, such as regret minimization, are known to converge to an equilibrium (Reference: Regret Minimization in Games with Incomplete Information. NeurIPS 2007), our analysis proves that the Nash Equilibrium in the LCM setting yields poor performance. Thus, learning provides no escape from this inefficient outcome.
>
> **In contrast, PVM is inherently robust in a repeated-game setting.** Its DSIC property makes truthful reporting a dominant strategy—the optimal action for any platform regardless of other platforms' strategies. Since this strategy is unconditionally optimal in each round, its optimality extends directly to the repeated game, eliminating any incentive for long-term strategic deviation.
>
> ## Q3: PVM Threshold Implementation
>
> The PVM mechanism is designed to be highly efficient, with threshold computation handled offline to ensure low-latency online performance.
>
> **(1) Offline Precomputation:**   While a general implementation could theoretically require precomputing an exponential number of thresholds ( $O(2^n)$ ), the equilibrium analysis central to our paper simplifies this significantly. For our use case, **only $n$ thresholds are needed**---one for each platform. This makes the offline precomputation step fast and practical for any realistic number of advertising platforms.
>
> **(2) Online Runtime Complexity:**  The online attribution process is also very efficient. To make a decision for a single conversion, the mechanism performs a check for each of the $n$ platforms. Each individual check requires finding the maximum timestamp among the other $n-1$ platforms (an $\mathcal{O}(n)$ operation) and a single comparison. Therefore, the **total online runtime complexity per conversion event is $\mathcal{O}(n^2)$**.
>
> But ad attribution is typically an offline batch process without stringent latency requirement.
>
> ## Q4: Advertiser Utility
> Our current objective is to maximize accuracy and fairness, assuming that a truthful signal is the most valuable input for an advertiser's downstream goal of optimizing ROI. Our work focuses on creating this foundational layer of truthful reporting.
>
> We agree that the idea of a multi-level game involving strategic interplay—such as strategic time reporting by platforms and strategic attribution by advertisers—is fascinating. **A full, joint-optimization game that models both advertiser and platform utilities while integrating the attribution and budget allocation processes represents a compelling direction for future research.** This convinces us that the field of ad attribution is rich with open questions, and we hope our foundational work can serve as a catalyst for such studies.
>
> ## Q5: Noise Robustness
> PVM provides a robust foundation upon which engineering solutions can be built to address these issues:
> - **Jittered Impressions:** Under a mechanism like LCM, platforms introduce large, minute-scale strategic delays to alter the click sequence. PVM’s DSIC property eliminates these strategic errors. The only remaining temporal noise is millisecond-level “jitter” from network latency. This minor, random noise is highly unlikely to change which platform was the true last click, making PVM's outcome robust to this issue.
> - **Delayed Reports:** We interpret this concern as the risk that operational delays could corrupt data and lead to inaccurate estimates of the click-time distributions that PVM relies on. This, in fact, highlights a key advantage of our design. **PVM’s DSIC property incentivizes platforms to eliminate strategic delays, ensuring the collected data is fundamentally honest.** This provides a clean baseline. Over time, an advertiser can statistically model and filter out these non-strategic latencies to better estimate the true distributions. This creates **a positive feedback loop**: better $F_i$ estimates lead to more precise PVM thresholds. Such a learning process is impossible under LCM, where strategic and operational delays are indistinguishable.
> - **Soft Conversions:** This is an excellent point and a standard challenge in practical attribution.  We view it as an issue of business logic that must be defined before the mechanism is applied. An advertiser must first establish a clear, consistent rule for what event constitutes the conversion $t_0$. PVM's logic can be readily applied to any consistently defined conversion event.

---

> > ### Comment · Reviewer_Fr9B · 2025-08-01
> > **Remaining Concern: Strategic Feasibility of Timestamp Manipulation**
> >
> > Thank you for the detailed and well-articulated rebuttal. I appreciate the authors’ thoughtful engagement with the concerns raised, and I continue to find the paper technically solid, with a compelling mechanism design approach to a timely problem in ad attribution.
> >
> > ## Strengths Maintained:
> > The paper remains strong in its theoretical modeling and empirical evaluation. The proposed Peer-Validated Mechanism (PVM) is a well-motivated and dominant-strategy alternative to LCM, and the real-world data experiments are a valuable contribution.
> >
> > ## Addressed Concerns:
> > Q2 (Learning Dynamics), Q3 (Threshold Complexity), Q4 (Advertiser Utility), and Q5 (Noise Robustness) were all satisfactorily addressed in the rebuttal. The authors provided solid clarifications on DSIC in repeated settings, computational feasibility, and the practical robustness of PVM under noisy or delayed inputs.
> >
> > ## Remaining Concern: Strategic Feasibility of Timestamp Manipulation
> > While the authors offer a detailed narrative arguing that timestamp misreporting is feasible—especially in direct-to-app-store funnels—their rebuttal lacks citations or empirical evidence to support these central claims.
> > * It is plausible that in some settings (e.g., mobile install attribution post-ATT), advertisers rely on platform-reported timestamps and are limited in their ability to audit.
> > * However, industry-wide assumptions of unilateral platform control over timestamps are questionable. Many attribution setups (e.g., server-side logging, MMPs, SKAdNetwork, etc.) involve independent logging mechanisms or validation steps.
> > * Furthermore, regulatory and reputational risks make large-scale strategic manipulation unlikely for major platforms.
> > * The **absence of references, technical documentation, or any quantifiable scope** on when and how platforms can delay timestamp reporting **limits the credibility and generality of the authors’ model** (weaknesses W1 and W2).
> >
> > Thus, while I now better understand the **intended modeling scenario**, I remain unconvinced that it broadly reflects industry practice without further substantiation.
> >
> > ## Confidence and Recommendation:
> > I want to explicitly note that I am not an expert in game-theoretic mechanism design, and my evaluation reflects that limitation.
> > My confidence score remains at 2: I am willing to defend the review but do not have the background to strongly advocate for the paper’s acceptance.
> > Given the above, and especially the still-uncertain real-world feasibility of the core assumption, I will maintain my overall score of 4 (Borderline Accept).
> > * I believe the paper is technically solid and would be a valuable contribution if its assumptions are valid.
> > * But due to the speculative nature of the practical setting and my limited expertise, I am unlikely to fight for acceptance.
> >
> > If the authors provide references or a more rigorous treatment of the empirical context (e.g., how common misreporting truly is), I would reconsider this score.

---

> > > ### Author Response · Authors · 2025-08-07
> > >
> > > Dear Reviewer Fr9B,
> > >
> > > We hope this message finds you well. As the rebuttal phase is nearing its end (with less than two days remaining), we would like to kindly follow up regarding our response to your comments. We would greatly appreciate it if you could let us know whether our reply has addressed your concerns adequately, or if there are any remaining questions or clarifications you'd like us to provide.
> > >
> > > Thank you again for your time and effort in reviewing our work.
> > >
> > > Best regards,
> > >
> > > Authors

---

> ### Author Response · Authors · 2025-08-02
>
> We sincerely thank you for the active feedback throughout the review process! As members of the industry, we have encountered the challenges discussed here in practice. The concerns we raise are not hypothetical, but based on issues we have observed in real-world deployment. To support this, we have gathered relevant academic papers, technical documentation, and reports from leading MMPs to provide empirical grounding for our model.
> ## 1. References for Empirical Context
> ### 1.1 References for The Drawbacks of Redirection-Based Tracking
> - **User Experience**:
>   - Netravali, R., et al., Vesper: Measuring Time-to-Interactivity for Web Pages, NSDI ’18
>   - Bocchi, E., et al., Measuring the Quality of Experience of Web Users, ACM SIGCOMM
> - **Security Vulnerabilities**:
>   - Somé, D. F., 2019, Tracking the Trackers: A Large-Scale Analysis of Web Tracking, WWW ’19
> ### 1.2 References for Industry-wide non-redirect tracking
> The technical foundation for our model is the "Parallel Tracking" architecture, now a standard across the industry.  As **Google's official documentation** describes,
>
> *"With parallel tracking, customers are delivered directly to your landing page while click measurement happens in the background. Here’s what parallel tracking looks like:
> (1)Customer clicks your ad.
> (2)Customer sees your landing page.
> (3)At the same time, in the background: Google Ads click tracker loads; tracking URL loads; if you use more than one click tracker, additional redirects may load.”*
>
> Leading platforms have since **mandated** this approach:
> - **Google Ads** required Parallel Tracking for Search, Shopping, and Display campaigns as of October 30, 2018, and extended to Video campaigns on April 30, 2021. **[Google Apps Help: About Parallel Tracking]**
> - **Microsoft Ad** mandated it for all new accounts created after May 31, 2020, and completed a forced migration of existing accounts by mid Jan 2021. **[Microsoft Advertising: AccountProperty Data Object - Campaign Management]**
>
> Parallel Tracking decouples click measurement from navigation by eliminating the user-facing redirect chain. This allows a platform to postpone its background reporting call, meaning the timestamp recorded by a third-party tracker is no longer independent but is instead determined by the platform’s reporting delay—all without any perceptible impact on the user’s experience.
> ### 1.3. References for advertiser-uncontrolled landing page
> To conservatively estimate the scale of ads ending on uncontrollable landing pages, we look at just two app stores. According to 2024 data from **Business of Apps**, Google Play and iOS saw 137.8 billion downloads. With  **ASO Mobile** reporting that 16% are ad-driven, this suggests over 22 billion conversions happened on app store pages. And this is still an underestimate, as it excludes major platforms like Amazon and TikTok Shop.
> ## 2. Why server-side logs, MMPs, and SKAdNetwork don’t solve the gap?
> - **Server-side logs**: It is viable only when the advertiser controls the landing page—impossible for App Store, Amazon, etc.
> - **MMPs** ：The shift to redirectless tracking and privacy policies has undermined MMPs’ ability to independently verify user-level attribution. Leading MMPs like Adjust are now learning to work with aggregate data instead of tracking users directly. **[Adjust:What is the identifier for advertisers (IDFA)?]**
> - **SKAdNetwork**: SKAdNetwork introduces a random delay and returns only aggregated post-backs . Thus most iOS advertisers still rely on platform-reported click to optimise spend.**[ Economic Impact of Opt-in versus Opt-out Requirementsfor Personal Data Usage:The Case of Apple’s App Tracking Transparency (ATT)Lennart Kraft et al.]**
>
> ## 3. Regulatory and Reputational Risks  for Major Platforms?
> First, we’d like to highlight a key point: while legal rules and after-the-fact remedies can help, any existing vulnerability may still be exploited. We believe a better solution is mechanism design, which removes the incentive to cheat from the start.
>
> Ad attribution has long been a target for fraud, as shown in academic and industry reports[**Click Fraud in Digital Advertising: A Comprehensive Survey, Computers 2021; Appsflyer:  Glossary/Mobile Ad Fraud**].
>
> For example, as Reviewer nsc8 noted, click spamming in redirect-based tracking involved sending many fake clicks after one real one—something MMPs could often detect. In redirect-free settings, fraudsters can just delay a real click, leaving no pattern and making detection much harder. **This turns detection into an expensive arms race.** Our PVM avoids that: as rewards depend on peer reports rather than its own, spamming or delaying has no benefit.
>
> While regulation and reputation may limit manipulation in some markets, this can‘t be assumed globally. In regions with weaker enforcement—such as parts of Asia or Latin America—platforms often face fewer constraints, making such tactics more likely. **[Anura: Surprising Ad Fraud Statistics]**

---

> ### Author Response · Authors · 2025-08-09
>
> Apologies for the additional message, but we wanted to follow up since we haven't received specific feedback on our further clarifications yet, and we are unsure if we fully addressed your concerns.
>
> To further clarify **the generalizability of our model’s time-delay strategy**, the assumption that **'Advertisers interact with a small number of platforms'**, and the potential impact of **reputational concerns on large platforms**, we are providing these additional clarifications.
>
> Our intention is solely to address any remaining doubts, not to pressure you into adjusting your initial positive rating, which we fully understand and appreciate. Thank you!
>
> ## (1) Generalizability of Our Model’s Time-delay Strategy
> While direct academic papers on time-delay manipulation are scarce, as we have stated before, the technical feasibility is grounded in industry-wide practices such as parallel tracking, which allows platforms like Google to emit measurement timestamps decoupled from user navigation ([Google Ads Help, 2021][1]; [Microsoft Advertising Docs, 2021][2]). This gives platforms the ability to control the click report timing.
> **Additionally, click spamming and click hijacking—both prevalent fraud techniques—are analogous to time-delay manipulation([3][4][5]).**
> - **Click spamming** involves generating real clicks followed by fraudulent ones to inflate recorded clicks and steal attribution. Studies show this is common in fraud schemes.
> - **Click hijacking** allows manipulation without a real click by intercepting user data (via cookies or scripts) and sending fake click signals. This tactic has been widely observed in mobile ad networks.
>
> While obtaining the necessary user data for click hijacking has become increasingly difficult due to privacy regulations and security protections, this only highlights that platforms still have the **technical capability** and **incentive** to engage in such behavior when they have access to this data. **When combined with real click data, the ability to manufacture fraudulent clicks equates to the manipulation described in our model.**
>
> ## (2) Reference for "Advertisers interact with a small number of platforms"
> AppsFlyer's 2015 case study shows that multi-touch installs typically involve an average of 2.7 sources ([AppsFlyer, 2015][6]). In practice, markets such as the US and China are dominated by a small number of major platforms:
> - US: Google Ads, Meta Ads (Facebook/Instagram), Amazon Ads, TikTok Ads
> - China: Tencent Ads, ByteDance Ads, Alibaba Ads, Baidu Ads
>
> This supports the assumption that n ≤ 5 is realistic.
> ## (3) Reputational Concerns on Large Platforms
> We fully agree that regulatory and reputational pressures are strong deterrents, especially for major platforms. However, as highlighted by the **Nash equilibrium**, only equilibrium strategies are stable in the long run.
>
> This means that **once a platform—whether a smaller player or one exploiting a regulatory loophole—starts misreporting, others may follow, not out of desire, but to avoid severe performance losses or competitive disadvantages**. In such a scenario, truthful reporting becomes unsustainable, despite initial preferences for honesty.
>
> We’ve even seen reputable platforms express frustration, asking: **“What can I do when others are stealing my credit?”** Unfortunately, in this competitive environment, misreporting becomes the only rational choice to remain competitive.
>
> Furthermore, studies show that even large platforms are not immune to fraud. For example, the **Anura Fraud Statistics (2024)** report highlights that affiliate marketing and programmatic advertising experience fraud rates of 45% and 50%, respectively, with **even major platforms like Google and Facebook facing fraud rates of 5%-10%** ([Anura, 2024][7]).
>
> ## References
>
> [1] Google Ads Help. (2021). About Parallel Tracking.
>
> [2] Microsoft Advertising Docs. (2021). Parallel Tracking for Campaigns.
>
> [3] Sadeghpour, S., & Vlajic, N. (2021). Click fraud in digital advertising: A comprehensive survey. *Computers*, 10(12), 164.
>
> [4] DataDome. (2024). Click spamming: What it is & how to prevent it.
>
> [5] O'Neill, M., & Jang, J. (2012). Clickjacking: Attacks and defenses. *Proceedings of the USENIX Security Symposium*, 1-15.
>
> [6] AppsFlyer. (2015). *Case Study: Retail*.
>
> [7] Anura. (2024). *Ad Fraud Statistics*.
>
>
> **Finally, we would like to express our gratitude for your suggestions and questions, which have been extremely helpful to us!**

---

### Author Response · Authors · 2025-08-09

Dear Reviewers,

Thank you once again for your thoughtful and constructive feedback. We greatly appreciate the time and effort you have invested in reviewing our paper and providing valuable insights!

We are pleased to have addressed all of your concerns during the rebuttal phase, and we are grateful for the positive feedback you have given after the revisions. We will carefully revise the paper to ensure that all of your suggestions are fully incorporated, further strengthening the clarity and impact of our work.

We would like to emphasize that our paper introduces the **first game-theoretic model for ad attribution**, addressing the challenge emerging from the **redirect-less paradigm**—where platforms can strategically delay timestamps to affect attribution outcomes. We propose **the Peer-Validated Mechanism (PVM)**, a novel DSIC mechanism that ensures platforms cannot steal attribution through timestamp delays. Our theoretical analysis shows that PVM **outperforms the Last-Click Mechanism (LCM)** in both accuracy and fairness. In homogeneous settings, PVM is the **optimal DSIC mechanism** with the highest  accuracy; and in all settings, PVM achieves **optimal fairness**. We further validate these properties through empirical experiments using real-world click data.

The core novelty of our work lies in introducing **a completely new perspective** to ad attribution: treating it as a mechanism design problem in which advertising platforms are strategic agents with their own incentives. This shift moves the focus from detecting and correcting manipulation after the fact to designing mechanisms where **manipulation is not incentive-compatible in the first place**. Our Peer-Validated Mechanism is the first DSIC mechanism for ad attribution, with proven advantages in both accuracy and fairness, and we believe it can serve as **a starting point** for a broader research agenda—extending to repeated-game settings, integrating with bidding strategies, and inspiring new applications of mechanism design in online advertising.

We sincerely hope that the paper will be accepted by NeurIPS so that more researchers in the community can engage with this perspective and further advance research in this important field!

Best regards,

Authors

---

### Note · Authors · 2025-08-13

Dear Area Chair and Reviewers,

We are immensely grateful for the rigorous and constructive discussion throughout the review process. The reviewers’ insightful feedback has been invaluable in strengthening the paper, and we will carefully incorporate their suggestions into the final version.

Our work is **the first to apply a game-theoretic framework to ad attribution**, tackling the strategic timestamp manipulation that plagues the industry. We rigorously prove that the standard Last-Click Mechanism (LCM) is not incentive-compatible (DSIC) and is deeply flawed in accuracy and fairness. In its place, we propose the Peer-Validated Mechanism (PVM), a novel DSIC solution praised by reviewers as "simple and optimal". Its optimality in homogeneous settings was noted as a "surprising and novel" technical result, and its superiority is robustly validated on real-world data.

We are pleased that the rebuttal **successfully resolved all concerns, leading reviewers who initially held reservations to raise their scores**. Crucially, the discussion solidified the practical premise of our work: the opportunity for strategic manipulation is a large-scale, urgent issue in the modern "redirect-less" advertising ecosystem, where traditional, redirection-based verification is no longer viable for the many campaigns with advertiser-uncontrolled landing pages (e.g., app-installs,e-commerce campaigns ).

This urgent, real-world problem exposes a deeper, foundational flaw in how attribution has been traditionally conceived. For too long, the field has treated platforms as passive data providers, not strategic agents, leading to a costly and inefficient 'arms race' of fraud versus detection. Our work argues for **a paradigm shift in thinking: from detection to design**. By making honesty the most profitable strategy, PVM's DSIC property eliminates the incentive for manipulation at the source, rendering tactics like click spamming or strategic delays ineffective for a rational agent. This provides a fundamentally more elegant and robust solution.

We believe our paper offers more than just a new mechanism; it provides a new, more robust framework for thinking about trust and incentives in computational advertising. We are confident in this contribution and sincerely hope it will be shared with the NeurIPS community.

Best regards,

Authors

---

### Decision · Program_Chairs · 2025-09-17

**Decision:**

Accept (poster)

**Comment:**

This paper studies the strategic aspects of the last click attribution mechanism and shows a new mechanism that behaves much better when players are strategic and can misreport the click time to their advantage. The authors engaged with the reviewers to clarify some misconceptions, and the scores are overall positive.